# Automatically Learning Hybrid Digital Twins of Dynamical Systems

**Samuel Holt,*  Tennison Liu* & Mihaela van der Schaar**
DAMTP, University of Cambridge
Cambridge, UK
{sih31, tl522, mv472}@cam.ac.uk

## Abstract

Digital Twins (DTs) are computational models that simulate the states and temporal dynamics of real-world systems, playing a crucial role in prediction, understanding, and decision-making across diverse domains. However, existing approaches to DTs often struggle to generalize to unseen conditions in data-scarce settings, a crucial requirement for such models. To address these limitations, our work begins by establishing the essential desiderata for effective DTs. Hybrid Digital Twins (**HDTwins**) represent a promising approach to address these requirements, modeling systems using a composition of both mechanistic and neural components. This hybrid architecture simultaneously leverages (partial) domain knowledge and neural network expressiveness to enhance generalization, with its modular design facilitating improved evolvability. While existing hybrid models rely on expert-specified architectures with only parameters optimized on data, *automatically* specifying and optimizing HDTwins remains intractable due to the complex search space and the need for flexible integration of domain priors. To overcome this complexity, we propose an evolutionary algorithm (**HDTwinGen**) that employs Large Language Models (LLMs) to autonomously propose, evaluate, and optimize HDTwins.[2] Specifically, LLMs iteratively generate novel model specifications, while offline tools are employed to optimize emitted parameters. Correspondingly, proposed models are evaluated and evolved based on targeted feedback, enabling the discovery of increasingly effective hybrid models. Our empirical results reveal that HDTwinGen produces generalizable, sample-efficient, and evolvable models, significantly advancing DTs' efficacy in real-world applications.

## 1  Introduction

*Digital Twins* (DTs) are computational models that accurately simulate the states and temporal dynamics of real-world systems [1, 2]. They are particularly useful in modeling *dynamical systems*, which consist of multiple interdependent components that evolve over time [3, 4]. Take, for example, the epidemiological dynamics of a contagious disease containing various components, including infection rates, recovery rates, population movement, and intervention strategies. DTs can integrate these factors to simulate future outcomes (e.g. predict disease spread), understand system changes (e.g. examining shifts in disease dynamics for varying demographics), and evaluate the impact of control measures (e.g. to curb disease transmission) [5, 6].

**Desiderata.** A notable differentiator between DTs and general machine learning (ML) models is the emphasis on generalization. DTs are designed to simulate completely *unseen* scenarios or interventions at inference time. Therefore, a crucial consideration is ▶ **[P1] out-of-distribution**

---

*Equal contributions; authors listed in randomized order.
[2]Code is available at https://github.com/samholt/HDTwinGen.

**generalization:** the ability to generalize to state-action distributions beyond those observed during training. This challenge is often compounded by the scarcity of observational data available to accurately learn dynamics, highlighting the importance of ▶ **[P2] sample-efficient learning**. Additionally, the model should be ▶ **[P3] evolvable:** capable of efficiently adapting (i.e. with minimal retraining) to changes in the *underlying system dynamics*. This is particularly crucial in healthcare domains, such as epidemiological modeling and treatment planning, where DTs are regularly updated to reflect fundamental changes in disease transmission patterns (caused by viral mutations, vaccination coverage) or evolving drug resistance mechanisms, often with minimal additional data of emergent dynamics [6, 7].

Existing approaches for creating DTs primarily utilize two approaches: *mechanistic* models or ML-based *neural* models. Mechanistic models, denoted as $f_{\text{mech}}$, are closed-form equations grounded in domain knowledge such as biological or physical principles. They offer high accuracy and generalization given *sufficient* domain understanding but are limited in their ability to model systems where scientific knowledge is incomplete [8, 9]. Of related note, techniques have been introduced to discover governing equations directly from data, but face challenges in scaling to more complex problem settings [10, 11]. Conversely, neural approaches, $f_{\text{neural}}$, leverage neural networks (NN) to learn DTs directly from data, often requiring minimal knowledge [12, 13, 14, 15]. Such models are effective given *sufficient* training data that provides adequate coverage of state-action distributions, but struggle in data-scarce settings and are difficult to evolve to reflect changing conditions due to their overparameterized, monolithic nature.

**Key considerations.** Informed by this context, *Hybrid Digtal Twins* (**HDTwins**) combine the strengths of both approaches through compositions of neural and mechanistic components, i.e. $f = f_{\text{mech}} \circ f_{\text{neural}}$. Here, $f_{\text{mech}}$ symbolically incorporates domain-grounded priors, improving generalization and regularization while simplifying the complexity of patterns that have to be learned by the neural component. In other terms, $f_{\text{neural}}$ complements the mechanistic component by modeling complex temporal patterns in regions where the mechanistic model might be oversimplified or incomplete. Consequently, HDTwins can more accurately and robustly capture system dynamics, particularly in settings with *(limited) empirical data and (partial) domain knowledge*.

Conceptually, hybrid modeling involves two stages: model *specification*, determining the model structure (e.g. neural architecture, symbolic equations), and model *parameterization*, estimating model parameters (e.g. neural weights, coefficients). This process, with model specification in particular, has traditionally relied heavily on human expertise to craft problem-specific models [16, 17, 18, 19]. In this work, we investigate the feasibility of *automatically* designing hybrid models with minimal expert involvement, which would significantly enhance the efficiency and scalability of model development. This task is challenging, as it requires searching for optimal specification and corresponding parameters within a vast combinatorial model space [20, 21]. To address this, we introduce **HDTwinGen**, a novel evolutionary framework that autonomously and efficiently designs HDTwins. At a high level, our method represents hybrid model specifications in code and leverages large language models (LLMs) for their domain knowledge, contextual understanding, and learning capabilities to propose symbolically represented models and search the model space [22, 23, 24]. This is coupled with offline optimization tools to empirically estimate model parameters from training data. More specifically, HDTwinGen utilizes two LLM agents: the *modeling agent*, whose task is to generate novel model specifications, and the *evaluation agent*, which analyzes performance and provides targeted recommendations for improvement. Through multiple iterations, HDTwinGen efficiently evolves better performing hybrid models with informed modifications.

**Contributions:** ① *Conceptually*, we present the first work in *automated hybrid model design*, jointly optimizing model specification and parameterization of hybrid digital twins. ② *Technically*, we introduce **HDTwinGen**, a novel evolutionary framework employing LLMs and offline optimization tools to propose, evaluate, and iteratively enhance hybrid models. ③ *Empirically*, we demonstrate that our method learns more accurate DTs, achieving ▶ better out-of-distribution generalization, ▶ sample-efficient learning, and ▶ increased flexibility for modular evolvability.

## 2 Digital Twins of Dynamical Systems

A dynamical system $\mathcal{S} := (\mathcal{X}, \mathcal{U}, \Phi)$ is a tuple of its $d_{\mathcal{X}}$-dimensional state space $\mathcal{X} \subseteq \mathbb{R}^{d_{\mathcal{X}}}$, an (optional) $d_{\mathcal{U}}$-dimensional action space $\mathcal{U} \subseteq \mathbb{R}^{d_{\mathcal{U}}}$, and a dynamics model $\Phi$. The state at time

$t \in \mathcal{T} \subseteq \mathbb{R}_+$ is represented as a vector, $x(t) \in \mathcal{X}$ and similarly the action taken is represented as a vector $u(t) \in \mathcal{U}$. The *continuous-time* dynamics of the system can be described by $dx(t)/dt = \Phi(x(t), u(t), t)$, where $\Phi : \mathcal{X} \times \mathcal{U} \times \mathcal{T} \to \mathcal{X}$. We optionally consider the existence of some policy $\pi : \mathcal{X} \to P(\mathcal{U})$ that acts on the system by mapping a state $x(t)$ to a distribution over actions $u(t)$.

**Digital Twins.** Digital twins (DTs) aim to approximate $\Phi : \mathcal{X} \times \mathcal{U} \times \mathcal{T} \to \mathcal{X}$ using a computational model $f_{\theta, \omega(\theta)} \in \mathcal{F}$ learned from data. Here, we use $\theta \in \Theta$ to denote the specification of the model (e.g. linear) and $\omega(\theta) \in \Omega(\theta)$ to indicate the set of parameters specified by $\theta$. Additionally, $\mathcal{F}$, $\Theta$, and $\Omega(\theta)$ are the spaces of all possible models, specifications, and parameters, respectively. Next, we outline the key desiderata for a DT:

**[P1] Generalization to unseen state-action distributions.** As DTs are required to simulate varying conditions, they should extrapolate to state-action distributions not observed during training time. Formally, the generalization error $\mathbb{E}_{(x(t), u(t), y(t)) \sim p_{OOD}}[\mathcal{L}(f_{\theta, \omega(\theta)}(x(t), u(t)), y(t))]$ should be minimized, where $\mathcal{L}$ is some loss function, and $p_{OOD}$ represents the out-of-distribution scenario.
**[P2] Sample-efficient learning.** Given the often limited availability of real-world data, DTs should learn robustly from minimal empirical data. In other words, they must have good *sample complexity*, achieving the desired level of generalization with a limited number of observations [25].
**[P3] Evolvability.** Dynamical systems are, by nature, non-stationary and evolve over time [26, 27]. From a modeling perspective, the DT should be easily evolved to reflect changing underlying dynamics, minimizing the need for additional data or expensive model re-development, i.e. $\theta$ and $\omega(\theta)$ should be easily adjustable to reflect changing system dynamics.

For the purpose of model learning, we assume access to an offline dataset containing $N \in \mathbb{N}^+$ trajectories, where the measurements of the systems are made at discrete time points $[T] = [t_1, t_2, \ldots T]$. This dataset, $\mathcal{D} = \{\{(x^{(n)}(t), u^{(n)}(t), y^{(n)}(t)) \mid t \in [T]\}\}_{n=1}^N$, contains state-action trajectories sampled regularly over time, where $y^{(n)}(t) = x^{(n)}(t + \Delta t)$ represents the subsequent state.

# 3 Hybrid Digital Twins

**HDTwin.** A Hybrid Digital Twin is a composition of mechanistic and neural components, represented as $f_{\theta, \omega(\theta)} = f_{\text{mech}} \circ f_{\text{neural}}$ [18, 28]. This class of hybrid models offers several advantages that align with our desiderata. The mechanistic component allows partial knowledge to be encoded through its symbolic form, which, while not sufficient alone to accurately predict underlying dynamics, is complemented by the neural components that learn from available data. This combination aids in generalization (**[P1]**), especially moving beyond conditions observed in training, and improves sample complexity (**[P2]**). Furthermore, the mechanistic component can be quickly and easily updated with new parameters due to its simpler, lower-dimensional structure, allowing the overall model to adapt efficiently to remain accurate in changing conditions (**[P3]**). In this work, we focus on *additive* compositions, $f_{\theta, \omega(\theta)} = f_{\text{mech}} + f_{\text{neural}}$, as they are more interpretable. Additionally, it enables individual contributions of mechanistic and neural components to be easily disentangled and simplifies the optimization to allow gradient-based methods [16]. Nonetheless, we encourage future works to investigate alternative composition strategies (e.g. branching composition) to develop more advanced HDTwins [29].

Learning the hybrid model can be decomposed into two steps: *(1) model specification*, or learning the structure, $\theta \in \Theta$, of the dynamics function that describes how the system evolves over time; and *(2) model parameterization*, which estimates the specific values of parameters $\omega(\theta) \in \Omega(\theta)$ for a given specification $\theta$. For instance, the logistic-growth model specifies a structure for population growth, while parameterization involves estimating the growth rate and carrying capacity.[3] More generally, this learning problem can be mathematically formulated as a bilevel optimization problem:

$$\theta^* = \arg\min_{\theta \in \Theta} \mathcal{L}_{\text{outer}}(\theta, \omega^*(\theta)), \quad \text{where} \quad \omega^*(\theta) = \arg\min_{\omega \in \Omega(\theta)} \mathcal{L}_{\text{inner}}(\theta, \omega(\theta)) \tag{1}$$

Here, the upper-level problem involves finding the optimal specification $\theta^*$ that minimizes the outer objective $\mathcal{L}_{\text{outer}}$, while the lower-level problem involves finding the optimal parameters $\omega^*(\theta)$ for a given specification $\theta$ that minimizes the inner objective function $\mathcal{L}_{\text{inner}}$. To be more concrete, the

---

[3]This model relates population size $N$, growth rate $r$, and carrying capacity $K$, $\partial N/\partial t = rN(1 - N/K)$ [30].

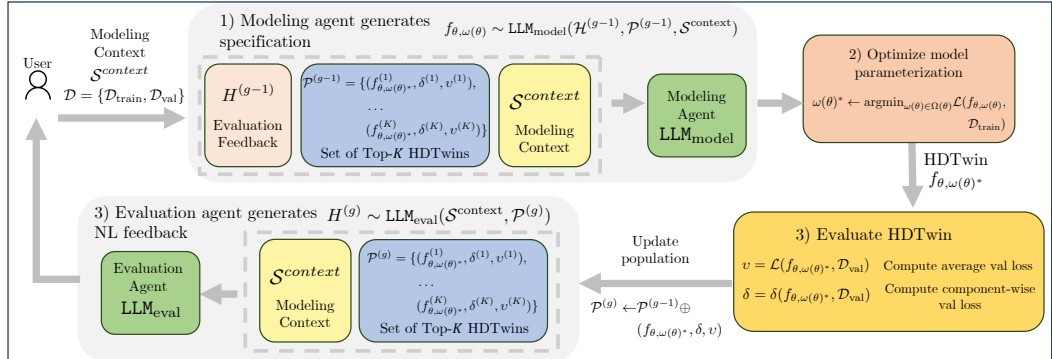

Figure 1: **HDTwinGen: evolutionary framework.** The process begins with user-provided modeling context $\mathcal{S}^{\text{context}}$ and $\mathcal{D} = \{\mathcal{D}_{\text{train}}, \mathcal{D}_{\text{val}}\}$. **1)** In iteration $g$, the *modeling agent* generates model specification as a Python program $f_{\theta,\omega(\theta)}$. **2)** Parameters are optimized using the *offline optimization tool* to yield $f_{\theta,\omega^*(\theta)}$. **3)** The HDTwin is evaluated based on model loss $\upsilon$ and component-wise loss $\delta$. Subsequently, the model pool $\mathcal{P}^{(g)}$ is updated with top-$K$ models. **4)** The *evaluation agent* provides targeted feedback for model improvement $H^{(g)}$ by analyzing models in $\mathcal{P}^{(g)}$ using performance metrics requirements outlined in $\mathcal{S}^{\text{context}}$. This iterative loop repeats for $G$ iterations.

outer objective measures the generalization performance, empirically measured on the validation set $\mathcal{L}_{\text{val}}$, while the inner objective measures the fitting error, as evaluated on the training set $\mathcal{L}_{\text{train}}$.

**Combinatorial search space.** The space of possible specifications $\Theta$ (e.g. different networks, functional forms) is discrete and combinatorially large, while $\Omega(\theta)$ represents the continuous space of parameters to be optimized. Selecting the optimal $\theta, \omega(\theta)$ thus involves searching through a vast combinatorial space. Performing this search through traditional means, such as genetic programming [21] or evolutionary algorithms [20], is computationally challenging, time-consuming, and often technically infeasible. To the best of our knowledge, our work is the first to address the problem of automatic HDTwin development, where we incorporate LLMs (combined with offline optimization tools) to automatically optimize both the specification and the parameterization of hybrid models.

## 4 HDTwinGen: Automatic Design of HDTwins

Human experts craft models by making strategic design decisions based on their domain knowledge, starting with a sensible initial model specification and performing intelligent modifications based on empirical evaluations. Our key insight is that LLMs can effectively emulate these capabilities to efficiently navigate the search space in Equation (1) and *autonomously design HDTwins*. More specifically, our method utilizes LLMs for three major purposes: ▶ **source of domain knowledge**, where LLMs inject domain-consistent knowledge into the model specification, particularly through the symbolic representation $f_{\text{mech}}$; ▶ **efficient search**, by making intelligent modifications to the specification to converge more efficiently on the optimal hypothesis; and ▶ **contextual understanding**, enabling the algorithm to incorporate task-specific context and targeted feedback for model improvement [22, 23, 24].

**Overview.** We operationalize this insight through **HDTwinGen**, an evolutionary algorithm that iteratively evolves a population of candidate solutions to automatically search for the best HDTwin. Our approach employs a framework comprising three key elements: *(1)* human experts provide an initial system description, modeling objectives, and requirements as a structured prompt; *(2)* a *modeling agent* proposes new model specifications, optimizes their parameters on a training dataset, and collects validation performance metrics; *(3)* an *evaluation agent* assesses the proposed models using both data-driven performance metrics and qualitative evaluations against expert-defined objectives and requirements. The agents communicate using natural language and a custom code format representing the HDTwin model, facilitating autonomous and iterative model enhancement. An overview of our method is presented in Figure 1, with pseudocode in Appendix E.1.

**Initial prompt design.** The optimization process begins with a human expert providing a structured prompt, referred to as the *modeling context* $\mathcal{S}^{\text{context}}$. This modeling context outlines the system description, modeling objectives $\mathcal{L}$, and requirements $\mathcal{R}$:

1. The **system description** semantically describes the system, including state and action variables, giving the algorithm the contextual understanding necessary for informed model development.
2. The **modeling objective** specifies *quantitative* performance requirements via a metric $\mathcal{L}$.
3. The **modeling requirements** $\mathcal{R}$ are *qualitative* and described in natural language, detailing aspects such as interpretability (e.g. fully mechanistic or hybrid model) and additional scientific knowledge (e.g. a log-linear relationship between variables).

In practice, $\mathcal{R}$ can incorporate various requirements, allowing for the design of both purely mechanistic and hybrid models, a flexibility that we demonstrate experimentally. The model is represented in Python, where purely mechanistic specifications are represented in native Python and neural components are represented using PyTorch [31]. Moreover, $\mathcal{S}^{\text{context}}$ includes a `skeleton code` to guide the synthesis of executable code in a predetermined format. For illustrative purposes, an example of $\mathcal{S}^{\text{context}}$ is provided in Appendix E.4.

**Evolutionary optimization overview.** Given $\mathcal{S}^{\text{context}}$ as input, HDTwinGen performs $G$ iterations of optimization, where $G \in \mathbb{N}^+$. The population of proposed HDTwins at iteration $g$ is represented as $\mathcal{P}^{(g)}$. Each iteration creates a new candidate model based on previously created models in $\mathcal{P}^{(g)}$ and feedback. Only the top $K$ models are retained after each iteration, except when $g < K$, in which case all generated models are kept, i.e. $\max_{g \in [G]} |\mathcal{P}^{(g)}| = K$. Each model in $\mathcal{P}^{(g)}$ is characterized by a tuple containing its model specification (represented symbolically through code) and validation metrics. After completing $G$ iterations, the model with the best validation performance in $\mathcal{P}^{(G)}$ is selected as the final model.

## 4.1 Modeling Agent

**Proposing HDTwins.** The goal of the modeling step is to propose novel HDTwins based on previously proposed models and feedback from the evaluation agent. Specifically, on the $g$-th iteration, the modeling agent takes as input ▶ $\mathcal{P}^{(g-1)}$: the set of top-$K$ previously generated models; ▶ $H^{(g-1)}$: the most recent feedback produced by the evaluation agent (where on the initial step, $g = 1$, both are empty, i.e., $H^{(0)} = \varnothing$, $\mathcal{P}^{(0)} = \varnothing$); and ▶ $\mathcal{S}^{\text{context}}$: the modeling context. The modeling agent generates a model specification $\theta$ using a predefined code format (i.e. `skeleton code`). By observing multiple previously best-performing models and their performances, the modeling agent can exploit this context as a rich form of in context-learning and evolve improved specifications in subsequent generations [22]. Each generated specification emits its corresponding parameters, $\omega(\theta)$ are fitted to the training set $\mathcal{D}_{\text{train}}$. More formally, we represent this generative procedure as $f_{\theta,\omega(\theta)} \sim \text{LLM}_{\text{model}}(H, \mathcal{P}^{(g)}, \mathcal{S}^{\text{context}})$.

**Model specification.** To generate model specifications, the modeling agent decomposes the system into a set of components, with each component describing the dynamics of a specific state variable. In other words, for a system with $d_{\mathcal{X}}$ state variables, there will be $d_{\mathcal{X}}$ components. Each component is characterized by its own set of inputs and a unique dynamics function that describes the dynamics of its associated state variable over time. This modular representation enables independent analysis and optimization of individual components. In cases where $\mathcal{R}$ specifies purely mechanistic equations, the component dynamics are entirely defined using closed-form equations. Conversely, in a hybrid model, the mechanistic equation can be augmented with a neural network (implemented in PyTorch) to model residuals (i.e. in an *additive* fashion). The choice between mechanistic and hybrid models is left to the user, balancing the trade-off between transparency and predictive performance. Concretely, the specification step involves 'filling in' the `skeleton code` with a detailed body of code, specifying the decomposition, and delineating each component's dynamics function as a separate code structure (for a generated HDTwin example, see Appendix I).

**Model optimization.** The generated specification emits $\omega(\theta)$, which are treated as placeholder values, and are then optimized against the training dataset. Specifically, we optimize the mean squared error for the parameters that minimize this loss as $\omega^*(\theta) = \arg\min_{\omega(\theta)} \mathcal{L}(f_{\theta,\omega(\theta)}, \mathcal{D}_{\text{train}})$. In this work, we consider $\omega(\theta)$ to be continuous variables, and as such, we optimize $\theta$ by stochastic gradient descent, using the Adam optimizer [32]. However, we note other optimization algorithms, such as black-box optimizers, could also be used (for more details, see Appendix F, Equation (5)). The parameter optimization step then yields the complete model, $f_{\theta,\omega^*(\theta)}$.

**Quantitative evaluation.** For each generated model, we evaluate them quantitatively. Specifically, we collect the validation mean squared error loss per component, which we denote as

$\delta = [\delta_1, \delta_2, \ldots, \delta_{d_{\mathcal{X}}}]$ (Appendix F, Equation (6)). We also compute the validation loss of the overall model as well as $\upsilon = \mathcal{L}(f_{\theta, \omega^*(\theta)}, \mathcal{D}_{\text{val}})$. Finally, the generated model and its validation losses are included in a tuple and added to the top-$K$ models $\mathcal{P}^{(g)} \leftarrow \mathcal{P}^{(g-1)} \oplus (f_{\theta, \omega^*(\theta)}, \delta, \upsilon)$, where $\mathcal{P}^{(g)}$ automatically removes the lowest performing models, and also only adds a new model to $\mathcal{P}^{(g)}$ if it is unique. We highlight that we consider the top-$K$ models only to apply *selection pressure*, such that only the best-performing models are considered when generating the next HDTwin [33].

### 4.1.1 Evaluation Agent

**Model evaluation.** The goal of the evaluation step is to reflect on the current set of top-$K$ models, $\mathcal{P}^{(g)}$ against requirements $\mathcal{R}$ and provide actionable and detailed feedback to the modeling agent for model improvement: $H^{(g)} \sim \text{LLM}_{\text{eval}}(\mathcal{R}, \mathcal{P}^{(g)})$. We note that $H^{(g)}$ is provided in natural language and can be viewed as a dense feedback signal, a notable distinction from traditional learning methods, where feedback often takes the form of simple scalar values, such as loss gradients or rewards. Leveraging natural language feedback allows the agent to *(1)* engage in comparative analysis, identifying effective specifications in $\mathcal{P}^{(g)}$ contributing to higher performance and discerning patterns common in less effective models, informing its suggestions for further model improvement; *(2)* qualitatively evaluate models against qualitative requirements $\mathcal{R}$—leveraging the LLM's capacity to reason about proposed HDTwins to reflect these requirements via model improvement feedback.

**Enhancing search.** By providing rich feedback to improve model specification, the evaluation and modeling agent collaborate to efficiently evolve high-performing models. Empirically, in Appendix J, we observe that the evaluation agent provides targeted and specific feedback, including component-specific suggestions, proposing alternative decompositions, removing parameters, or introducing non-linear terms. It is noteworthy that the feedback $H^{(g)}$, expressed flexibly in natural language, could easily be further enriched through direct human feedback. We demonstrate this *human-in-the-loop* capability by including expert feedback during the optimization process through $H^{(g)}$ and observed that it was integrated into newly generated HDTwins. Though further investigation is beyond the scope of this work, this demonstration highlights promising avenues for augmenting human-machine collaboration in the autonomous design of DTs.

## 5 Related Works

For an extended related work, refer to Appendix B. Our work focuses on autonomously learning DTs from data, with several relevant research strands:

**Neural sequence models.** ML approaches commonly address learning system dynamics as a sequential modeling problem. In these settings, $f_{\theta, \omega(\theta)}$ are typically black-box models, where $\theta \in \Theta$ is the NN architecture and $\omega(\theta)$ are its weights. Early models like Hidden Markov Models [34] and Kalman filters [35] made simplifying Markovian and linearity assumptions, later extended to nonlinear settings [36, 37]. Subsequent models, including recurrent neural networks [38], along with their advanced variants [39, 40, 41], introduced the capability to model longer-term dependencies. More recent advancements include attention mechanisms [42] and Transformer models [43], significantly improving the handling of long-term dependencies in sequence data. Another line of work, Neural Ordinary Differential Equations (NODE) [14, 44, 45], interprets neural network operations as differential equations. These methods have found utility in modeling a range of complex systems [46, 47, 48, 49]. While deep sequence models are proficient at capturing complex dynamics, they are heavily reliant on training data for generalization (**[P1, P2]**), and their monolithic and overparameterized structures limit evolvability (**[P3]**).

**Mechanistic (discovery) models.** Beyond purely neural approaches, another line of work aims to discover a system's governing equations directly from data. Here $\theta \in \Theta$ are closed-form equations and $\omega(\theta)$ are their parameters. These include symbolic regression techniques [10], Eureqa [50], SINDy [11], D-CODE [51, 52], among others [52, 53] that search for $\theta$ and $\omega(\theta)$ from data. These techniques struggle to scale to higher-dimensional settings and rely on experts to perform variable selection and define the function set and primitives available to the search algorithms.

**Hybrid models.** Recent efforts have also created hybrid models by integrating physical laws with neural models. Physics-informed neural networks [15, 54], and methods including Hamiltonian Neural Networks [55], Lagrangian Neural Networks [56] integrate structural priors of physical

systems to improve generalization. These techniques introduce specialized mechanisms to incorporate *precisely known* physical principles. Additionally, [57] integrates prior ODE/PDE knowledge into a hybrid model, using specialized regularization to penalize the neural component's information content. [58, 59] consider settings where an expert equation is known, but equation variables are latent and unobserved. Correspondingly, they employ two sets of latent variables: one governed by expert equations and another linked to neural components. [60] performs data augmentation by sampling out-of-distribution trajectories from expert models. While existing approaches rely on expert models to perform the hybrid model design, HDTwinGen is an automated approach to jointly optimize hybrid model specification and its parameters.

## 6 Experiments and Evaluation

In this section, we evaluate HDTwinGen and verify that it significantly outperforms state-of-the-art methods in modeling system dynamics over time from an observed dataset and corresponding system description.[4]

**Benchmark datasets**. We evaluate against **six** real-world complex system datasets; where each dataset is either a real-world dataset or has been sampled from an accurate simulator designed by human experts. Three are derived from a state-of-the-art biomedical Pharmacokinetic-Pharmacodynamic (PKPD) model of lung cancer tumor growth, used to simulate the combined effects of chemotherapy and radiotherapy in lung cancer [61] (Equation (2))—this has been extensively used by other works [62, 63, 64]. Here we use this bio-mathematical lung cancer model to create three variations of lung cancer under the effect of no treatments (**Lung Cancer**), chemotherapy only (**Lung Cancer (with Chemo.)**), and chemotherapy combined with radiotherapy (**Lung Cancer (with Chemo. & Radio.)**). We also compare against an accurate and complex COVID-19 epidemic agent-based simulator (**COVID-19**) [65], which is capable of modeling non-pharmaceutical interventions, such as physical distancing during a lockdown. Furthermore, we compare against an ecological model of a microcosm of algae, flagellate, and rotifer populations (**Plankton Microcosm**)—replicating an experimental three-species prey-predator system [66]. Moreover, we also compare against a real-world dataset of hare and lynx populations (**Hare-Lynx**), replicating predator-prey dynamics [67]. We detail all benchmark datasets details in Appendix C.

**Evaluation Metrics**. We employ mean squared error (MSE) to evaluate the benchmark methods on a held-out test dataset of state-action trajectories, denoted as $\mathcal{D}_{\text{test}}$, using the loss defined in Equation (5) and report this as $\mathcal{T}_{MSE}$. Each metric is averaged over ten runs with different random seeds, and we present these averages along with their 95% confidence intervals, further detailed in Appendix G.

**Benchmark methods**. To assess whether HDTwinGen is state-of-the-art, we compare it with the most competitive and popular neural network models, which, when modeling the dynamics of a system over time, becomes a form of ODE model, that is a neural ODE [14] with action inputs (**DyNODE**) [68]. Moreover, we also compare against a recurrent neural network (**RNN**) [69] and a state-of-the-art transformer (**Transformer**) [64]. We also compare against mechanistic dynamical equations derived from equation discovery methods for ODEs, including Genetic Programming (**GP**) [10] and Sparse Identification of Nonlinear Dynamics (**SINDy**) [11]. Lastly, we compare against a hybrid model (**APHYNITY**) that integrates prior knowledge in the form of ODEs into hybrid models, while penalizing the information content from the neural component [57]. Moreover, we compare against the ablations of our method, of the zero-shot generated HDTwin (**ZeroShot**) and this model with subsequently optimized parameters (**ZeroOptim**). We provide method implementation, hyperparameter, and experimental details in Appendix D.

## 7 Main Results

We evaluated all our benchmark methods across all our datasets tabulated in Table 1. HDTwinGen models the system the most accurately, achieving the lowest test prediction mean squared error on the held-out test dataset of state-action trajectories. In the interest of space, we include additional experimental evaluations in the appendix. Specifically, we also evaluate ▶ HDTwinGen performance on a suite of synthetically and procedurally generated benchmarks (Appendix H.9); ▶ comparisons

---

[4]Our implementation is available at https://github.com/samholt/HDTwinGen. We also provide a wider lab code repository at https://github.com/vanderschaarlab/HDTwinGen.

Table 1: **Benchmark method performance.** Reporting the test prediction MSE ($\mathcal{T}_{MSE}$) of the produced system models on held-out test datasets across all benchmark datasets. HDTwinGen achieves the lowest test prediction error. The results are averaged over ten random seeds, with $\pm$ indicating 95% confidence intervals.

| Method | Lung Cancer $\mathcal{T}_{MSE} \downarrow$ | Lung Cancer (with Chemo.) $\mathcal{T}_{MSE} \downarrow$ | Lung Cancer (with Chemo. & Radio.) $\mathcal{T}_{MSE} \downarrow$ | Hare-Lynx $\mathcal{T}_{MSE} \downarrow$ | Plankton Microcosm $\mathcal{T}_{MSE} \downarrow$ | COVID-19 $\mathcal{T}_{MSE} \downarrow$ |
|---|---|---|---|---|---|---|
| SINDy | 327±5.79 | 11.8±0.395 | 13.7±0.573 | 388±4.29e-14 | 0.00135±0 | 93.4±0.458 |
| GP | 158±94.1 | 154±505 | 171±8.99 | 514±381 | 0.00474±0.0564 | 10.1±18 |
| DyNODE | 327±5.8 | 52±47.1 | 16.3±5.58 | 439±0 | 0.00036±0.00078 | 74±2.36 |
| RNN | 1.17e+06±3.08e+04 | 708±86.1 | 136±5.6 | 3.71e+03±3.39e+03 | 0.0281±0.0406 | 1.38e+04±1.65e+03 |
| Transformer | 7.48±1.06 | 0.348±0.0618 | 0.216±0.0345 | 716±42.5 | 3.69e-05±1.83e-05 | **0.309±0.222** |
| APHYNITY | 9.06±1.37 | 81.6±81.3 | 1.21e+03±1.69e+03 | 321±12.6 | 4.21e-05±3.45e-05 | 88.8±9.97 |
| ZeroShot | 5.45e+03±6.71e+03 | 292±80.2 | 5.81e+03±4.02e+03 | 338±0 | 0.325±0.242 | 2.31e+03±2.24e+03 |
| ZeroOptim | 216±172 | 31.2±45 | 6.08±7.9 | 353±0 | 0.0132±0.00116 | 7.88±0.0414 |
| **HDTwinGen** | **4.41±8.07** | **0.0889±0.0453** | **0.131±0.198** | **291±30.3** | **2.51e-06±2.2e-06** | **1.72±2.28** |

Table 2: **Out of distribution shifts.** On a variation of the Lung Cancer (with Chemo. & Radio.), HDTwinGen is more robust to OOD shifts in unseen state-action distributions.

| Method | Lung Cancer (with Chemo. & Radio.) IID $\mathcal{T}_{MSE} \downarrow$ | OOD $\mathcal{T}_{MSE} \downarrow$ |
|---|---|---|
| DyNODE | 0.0115±0.0121 | 1.75±0.769 |
| SINDy | 0.302±0.286 | 5.9±2.55 |
| RNN | 1.43e+04±2.02e+03 | 1.84e+05±4.06e+04 |
| Transformer | 0.0262±0.00514 | 1.19e+04±2.78e+03 |
| ZeroShot | 4.95e+03±1.43e+04 | 1.91e+04±6.36e+04 |
| ZeroOptim | 3.49±0.0364 | 4.84±5.17 |
| **HDTwinGen** | **0.00872±0.0187** | **0.0846±0.0891** |

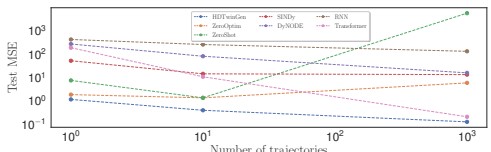

Figure 2: **Sample efficiency.** Analyzing performance as a function of the number of training trajectories in the Lung Cancer (with Chemo. & Radio.) dataset. We observe that HDTwinGen achieves the lowest test prediction error, even in the very challenging low data regime. This highlights the role of priors embedded in HDTwin in sample-efficient generalization.

against domain-specific baselines (Appendix H.8) and ► various ablation experiments, including ablation of LLM hyperparameters, prompt design, and algorithm settings (Appendices H.5 to H.7).

## 7.1 Insight Experiments

This section provides an in-depth analysis of HDTwinGen's effectiveness related to its benchmark counterparts. Specifically, we examine the core desiderata for an effective DT described in Section 2: **[P1]** out-of-distribution generalization, **[P2]** sample-efficient learning, and **[P3]** evolvability.

**[P1] Can an HDTwin generalize to out-of-distribution shifts?** To explore out-of-distribution shifts, we adapt the Lung Cancer (with Chemo. & Radio.) to produce a training dataset of states in a range that is outside those observed in the test set over all trajectories (Appendix H.1). We tabulate this in Table 2. Empirically, we find that HDTwinGen is more robust to out-of-distribution shifts than existing methods, benefiting from explicit decomposition and robust hybrid models. Notably, the neural network method DyNODE shows the largest relative error increase from IID to OOD by two orders of magnitude, while the mechanistic method SINDy exhibits a smaller increase by only one order of magnitude. This demonstrates the importance of hybrid models that leverage both neural and mechanistic components to enhance generalization performance under distribution shifts.

**[P2] Can HDTwinGen improve sample-efficiency in model learning?** To explore the low data settings, we re-ran all benchmark methods with fewer samples in their training dataset on the Lung Cancer (with Chemo. & Radio.) dataset. We plot this in Figure 2. Empirically, we observe that HDTwinGen can achieve lower performance errors, especially in lower-sample regimes.

**[P3] Can HDTwinGen evolve its modular HDTwin to fit the system?** We analyze this from an empirical point of view to determine if HDTwinGen can correctly evolve the generated HDTwin and reduce its prediction error over subsequent generations. We observe that HDTwinGen can indeed understand, reason, and iteratively *evolve* the generated code representation of the HDTwin to incorporate a better fitting HDTwin, as observed in Figure 3. In particular, the annotated results demonstrate that HDTwinGen effectively refines the hybrid model by strategically adjusting its neural and mechanistic components (in a fashion akin to human experts), leading to significant improvements

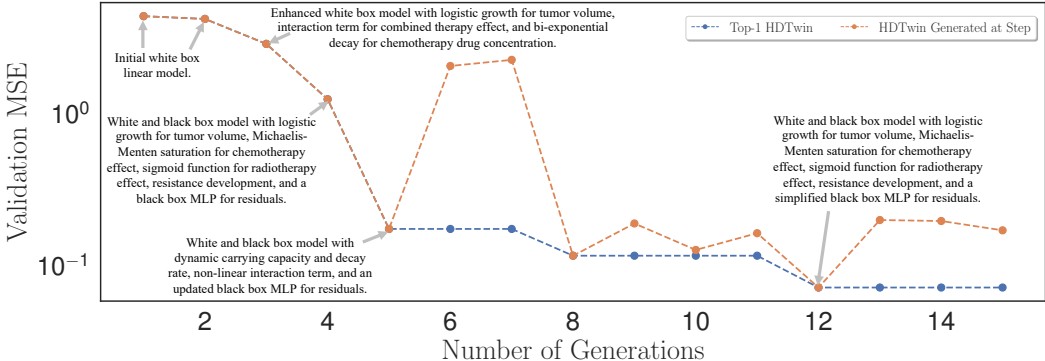

Figure 3: **HDTwinGen effectively evolves HDTwin.** Validation MSE of the HDTwin generated in each iteration, showing the Pareto-front of the best generated HDTwin (Top-1 HDTwin), and the generated HDTwin per generation step—additionally with a few of the HDTwins labeled with their model descriptions. HDTwinGen can efficiently understand, modify, and hence *evolve* the HDTwin to achieve a better-fitting model (Appendix H.4).

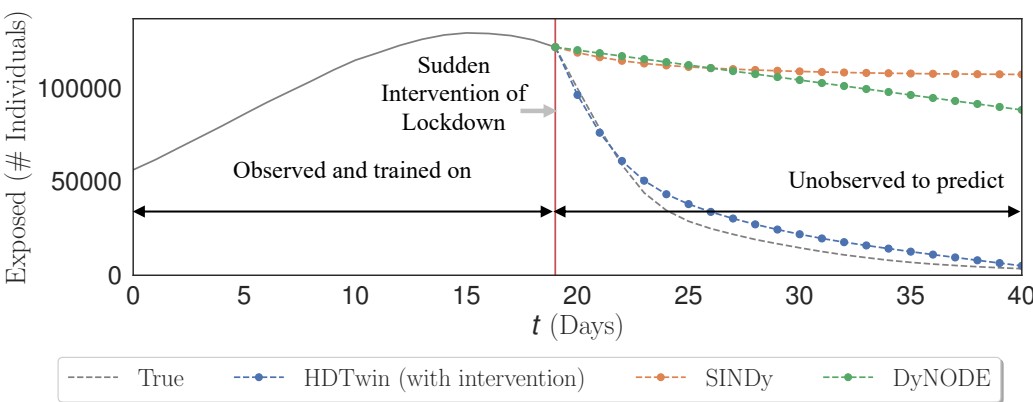

Figure 4: **COVID-19 unobserved intervention.** The symbolic code-based representation of HDTwin can be easily adapted to unobserved interventions through targeted adjustments of parameters.

in accuracy and robustness. This iterative evolution process demonstrates HDTwinGen's ability to adapt and optimize its modular components.

**Can HDTwinGen Understand and Modify Its HDTwin?** We investigate whether large language model (LLM) agents can take an optimized high-dimensional twin (HDTwin) from an existing benchmark dataset and adapt it to model an unobserved intervention that is not present in the training data. We note that this intervention emulates scenarios where the dynamics of the underlying system changes. We affirmatively answer this question by constructing a scenario where our COVID-19 simulator incorporates an unobserved intervention of a lockdown policy, which reduces physical interactions between individuals (Appendix H.2). As demonstrated in Figure 4, we observe that the code-model representation of the HDTwin can be (1) understood by the modeling agent LLM and (2) adapted in its parameters to accurately model and reflect this intervention. We find that HDTwinGen is the *only method capable of changing the overall functional behavior by modifying a single parameter in the model*; in contrast, all other existing data-driven methods require a dataset of state-action trajectories under the new dynamics introduced by this intervention.

**Ablation Studies**. We conducted ablation studies on HDTwinGen and found several key insights. First, retaining the top-$K$ models within the LLM context leads to improved model generation (Appendix H.5). Additionally, HDTwinGen is compatible with various LLMs and different temperature settings (Appendix H.6). It also benefits from including textual descriptions of the variables to be modeled as prior information (Appendix H.7). Finally, HDTwinGen can be specifically instructed to generate mechanistic white-box models if desired (Appendix H.10).

# 8 Limitations and Discussions

In summary, this work addresses the problem of learning digital twins for continuous-time dynamical systems. After establishing clear learning objectives and key requirements, we introduce Hybrid Digital Twins (**HDTwins**)—a promising approach that combines mechanistic understanding with neural architectures. HDTwins encode domain knowledge symbolically while leveraging neural networks for enhanced expressiveness. Conventional hybrid models, however, rely heavily on expert specification with learning limited to parameter optimization, constraining their scalability and applicability. To overcome these limitations, we propose a novel approach to *automatically* specify and parameterize HDTwins through **HDTwinGen**, an evolutionary framework that leverages LLMs to iteratively search for and optimize high-performing hybrid twins. Our empirical results demonstrate that evolved HDTwins consistently outperform existing approaches across multiple criteria, exhibiting superior out-of-distribution generalization, enhanced sample efficiency, and improved modular evolvability.

**Limitations.** While our results are promising, several important limitations remain. HDTwinGen's efficacy depends critically on human experts providing initial system specifications and on the underlying LLM's domain knowledge and model generation capabilities. Our current implementation focuses exclusively on continuous-time systems, which, although broadly applicable, represent only a subset of real-world systems. Future work could extend our approach through human-in-the-loop feedback mechanisms, integration with external tools, and expansion to broader system classes.

**Ethical implications.** We acknowledge the risk of bias transmission from the black-box LLMs into the evolved models. While our hybrid approach enables greater expert scrutiny through its human-interpretable components, we strongly recommend a comprehensive evaluation of evolved models for fairness, bias, and privacy concerns before deployment in sensitive applications.

## Acknowledgments and Disclosure of Funding

We thank the anonymous reviewers, area and program chairs, members of the van der Schaar lab, and Andrew Rashbass for many insightful comments and suggestions. TL and SH would like to acknowledge and thank AstraZeneca for their sponsorship and support. This work was supported by Microsoft's Accelerate Foundation Models Academic Research initiative.

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

# Appendix

## Table of Contents

# A HDTwinGen Overview

We provide an illustrative example of HDTwinGen working in practice in Figure 5.

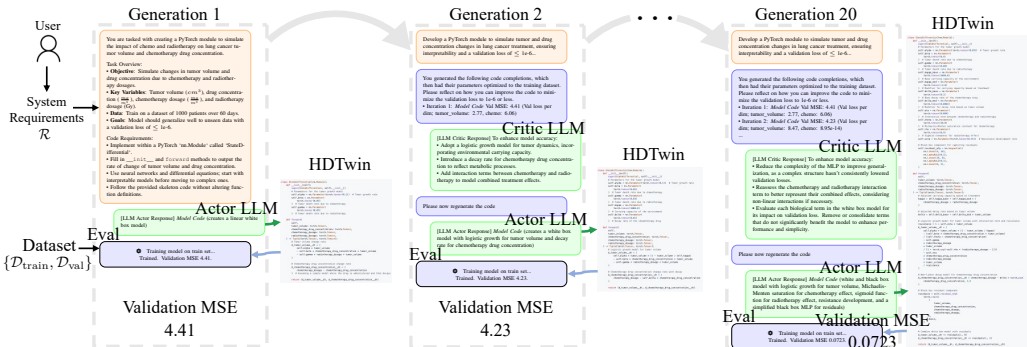

Figure 5: **HDTwinGen Illustrative Example in Operation.** HDTwinGen can generate and further evolve HDTwins for a particular system based on user-given system requirements and a dataset $\mathcal{D} = \{\mathcal{D}_{\text{train}}, \mathcal{D}_{\text{test}}\}$ of state-action trajectories. First, the system requirements—which include dataset statistics are incorporated into a prompt and fed into the modeling agent that returns the code for the HDTwin. This HDTwin is then trained on the training dataset $\mathcal{D}_{\text{train}}$, and a validation loss is computed with $\mathcal{D}_{\text{val}}$. In subsequent generations, the evaluation agent is given the existing generated top-$K$ HDTwins, their corresponding validation losses, and validation losses per component, and is asked to reflect on how to improve the HDTwin. This provides detailed, actionable feedback, leveraged from its inherent understanding, and provides this as detailed verbal feedback as $H$, whereby this feedback is next used with the modeling agent to generate the next HDTwin [P3]. This process iterates several generation times, and the best-performing HDTwin (w.r.t. validation performance) is returned. Overall, this produces an HDTwin that fulfills [P1-P3].

# B Extended Related Work

**Sequence models.** ML approaches frequently address system dynamics as a sequential modeling problem. Initial models like Hidden Markov Models [34] and Kalman filters [35] made simplifying Markovian and linearity assumptions, later extended to nonlinear settings [36, 37]. Subsequent models, including recurrent neural networks [38], along with their advanced variants [39, 40, 41], introduced the capability to model longer-term dependencies. More recent advancements include attention mechanisms [42] and transformer models [43], significantly improving the handling of long-term dependencies in sequence data. Another line of work, Neural Ordinary Differential Equations (NODE) [14, 44, 45], interprets neural network operations as integrations of differential equations to model continuous-time processes. Despite being initially driven by natural language processing applications [47], these methods have found utility in modeling complex systems like weather forecasting [46] and energy systems [48]. Furthermore, sequence models can be used in model-based RL [70].

**Physics-inspired models.** Beyond purely data-centric approaches, recent efforts have focused on integrating physical laws into neural system models. Physics-informed neural networks [15, 54] embed physical laws, often as partial differential equations, directly into the learning process. Other notable methods include Hamiltonian Neural Networks [55] and Lagrangian Neural Networks [56], which respect the structural principles of physical systems. These methods are primarily concerned with modeling *physics-related* phenomenon and require relatively precise knowledge about the system being modeled (e.g. specific differential equations or energy conservation principles) and specialized mechanisms to incorporate them. Regardless, they have demonstrated that the integration of known principles can significantly improve extrapolation abilities beyond the range of training data. We are similarly inspired to incorporate prior knowledge. In contrast, our work aims to integrate more general or partial knowledge flexibly into a hybrid model using LLMs within a evolutionary multi-agent framework, while introducing more generalized mechanisms to incorporate loose-form prior knowledge.

**Discovering closed-form models.** Closely aligned with our research are techniques aimed at discovering closed-form mathematical expressions from data. Symbolic regression [10, 71, 72] and methods like Eureqa [50], SINDy [11], and D-CODE [51, 52] have showcased their prowess in discovering physical laws from experimental observations. However, these techniques can struggle in higher-dimensional settings and rely on experts to perform the system decomposition to identify the most relevant variables before feeding this information to the algorithm. Additionally, they also rely on experts to specify the function set and mathematical operations that the algorithm uses to search for symbolic expressions. In contrast, our method autonomously learns both the system decomposition and the functional forms of component dynamics, potentially enhancing scalability and efficiency. Moreover, the incorporation of LLMs facilitates the flexible integration of prior knowledge at various stages of the search process [73, 74]. Furthermore, using LLMs to generate code, prior work has shown LLM multi-agent frameworks' ability to excel at large code-generation tasks [75, 76], which we could expect to apply here to scale up the size of the generated models in future works. Such future work could also explore acquiring features as well [77, 78].

## C Benchmark Dataset Environment Details

In the following, we outline the six real-world system dynamics datasets, where each dataset is either a real-world dataset or has been sampled from an accurate simulator designed by human experts.

### C.1 Cancer PKPD

Three of our environments that we sample a dataset from are derived from a state-of-the-art biomedical Pharmacokinetic-Pharmacodynamic (PKPD) model of lung cancer tumor growth, used to simulate the combined effects of chemotherapy and radiotherapy in lung cancer [61]—this has been extensively used by other works [62, 63, 64]. Here we use this bio-mathematical lung cancer model to create three variations of lung cancer under the effect of no treatments (**Lung Cancer**), chemotherapy only (**Lung Cancer (with Chemo.)**), and chemotherapy combined with radiotherapy (**Lung Cancer (with Chemo. & Radio.)**); for each model we sample a respective dataset. First, let us detail the general case of *Lung Cancer (with Chemo. & Radio.)*, which comes from the general model (Cancer PKPD Model), and then detail the variations.

**Cancer PKPD Model**. This is a state-of-the-art biomedical Pharmacokinetic-Pharmacodynamic (PKPD) model of tumor growth, that simulates the combined effects of chemotherapy and radiotherapy in lung cancer [61] (Equation (2))—this has been extensively used by other works [62, 63, 64]. Specifically, this models the volume of the tumor $x(t)$ for days $t$ after the cancer diagnosis—where the outcome is one-dimensional. The model has two binary treatments: (1) radiotherapy $u_t^r$ and (2) chemotherapy $u_t^c$.

$$\frac{dx(t)}{dt} = \Big( \underbrace{\rho \log\left(\frac{K}{x(t)}\right)}_{\text{Tumor growth}} - \underbrace{\beta_c C(t)}_{\text{Chemotherapy}} - \underbrace{(\alpha_r d(t) + \beta_r d(t)^2)}_{\text{Radiotherapy}} \Big)x(t) \tag{2}$$

Where the parameters $K, \rho, \beta_c, \alpha_r, \beta_r$ for each simulated patient are detailed in [61], which are also described in Table 3. Additionally, the chemotherapy drug concentration $c(t)$ follows an exponential

Table 3: **Cancer PKPD parameter values.**

| Model | Variable | Parameter | Parameter Value |
|---|---|---|---|
| Tumor growth | Growth parameter | $\rho$ | $7.00 \times 10^{-5}$ |
| | Carrying capacity | $K$ | 30 |
| Radiotherapy | Radio cell kill ($\alpha$) | $\alpha_r$ | 0.0398 |
| | Radio cell kill ($\beta$) | $\beta_r$ | Set s.t. $\alpha/\beta$=10 |
| Chemotherapy | Chemo cell kill | $\beta_c$ | 0.028 |

decay relationship with a half-life of one day:

$$\frac{dc(t)}{dt} = -0.5c(t) \tag{3}$$

where the chemotherapy binary action represents increasing the $c(t)$ concentration by $5.0\text{mg}/\text{m}^3$ of Vinblastine given at time $t$. Whereas the radiotherapy concentration $d(t)$ represents $2.0Gy$ fractions of radiotherapy given at timestep $t$, where Gy is the Gray ionizing radiation dose.

**Time-dependent confounding.** We introduce time-varying confounding into the data generation process. This is accomplished by characterizing the allocation of chemotherapy and radiotherapy as Bernoulli random variables. The associated probabilities, $p_c$ and $p_r$, are determined by the tumor diameter as follows:

$$p_c(t) = \sigma\left(\frac{\gamma_c}{D_{\max}}(\bar{D}(t) - \delta_c)\right) \qquad p_r(t) = \sigma\left(\frac{\gamma_r}{D_{\max}}(\bar{D}(t) - \delta_r)\right), \qquad (4)$$

where $D_{\max} = 13\text{cm}$ represents the largest tumor diameter, $\theta_c = \theta_r = D_{\max}/2$ and $\bar{D}(t)$ signifies the mean tumor diameter. The parameters $\gamma_c$ and $\gamma_r$ manage the extent of time-varying confounding. We use $\gamma_c = \gamma_r = 2$.

**Sampling datasets.** Using the above Cancer PKPD model, we sample $N = 1,000$ trajectories, which equates to $N = 10,000$ patients, where we sample their initial tumor volumes from a uniform distribution $x(0) \sim \mathcal{U}(0, 1149)$, and use the Cancer PKPD Equation (2) along with the action policy of Equation (4) to forward simulate patient trajectories for 60 days, using a Euler stepwise solver. This forms one dataset sample. We repeat this process with independent random seeds to generate $\mathcal{D}_{\text{train}}, \mathcal{D}_{\text{val}}, \mathcal{D}_{\text{test}}$. Specifically for each benchmark method run for random seed, we re-sample the datasets. For each variation described above, we either include the chemotherapy dosing action, chemotherapy and radiotherapy dosing action or neither. We further outline this dataset's system description and variable descriptions with the following prompt template as given in Appendix E.4.

## C.2 COVID-19

We use the accurate and complex epidemic agent-based simulator of COVASIM [65] to simulate COVID-19 epidemics. This is an advanced simulator that is capable of simulating non-pharmaceutical interventions (such as lockdowns through social distancing, and school closures) and pharmaceutical interventions (such as vaccinations). As this is an agent-based simulator, each agent is an individual in a population, and they can be in one of the following states minimally, of being susceptible to COVID-19, exposed, infectious or recovered (which includes deaths). We use the simulator with the default parameters set by the open source implementation of the simulator [5]. COVASIM runs a simulation for a population of individuals. To ensure an accurate simulation, we simulate 24 countries collecting trajectories for each, wherein each simulation we use a population size of $1,000,000$ individuals, and simulate each individual separately (disabling simulation rescaling) and start with a random number of individuals who are infected with COVID-19, $I(0) = \mathcal{U}(10,000, 100,000)$, and forward simulate the simulation for 60 days. We repeat this process with independent random seeds to generate $\mathcal{D}_{\text{train}}, \mathcal{D}_{\text{val}}, \mathcal{D}_{\text{test}}$. Specifically for each benchmark method run for random seed, we re-sample the datasets. We further outline this dataset's system description and variable descriptions with the following prompt template as given in Appendix E.4.

## C.3 Plankton Microcosm

This describes an ecological model of a microcosm of algae, flagellate, and rotifer populations, thus replicating an experimental three-species prey-predator system [66]. We use the dataset made available by [79][6]. The dataset consists of a single trajectory of 102 time steps, and we use a train, val, test split of 70%, 15% and 15%, ensuring the splits are along the time dimension to maintain the integrity of temporal causality, following their chronological order. We further outline this dataset's system description and variable descriptions with the following prompt template as given in Appendix E.4.

---

[5]COVASIM is an opensource simulator, from which we access it here https://github.com/InstituteforDiseaseModeling/covasim.

[6]The Plankton Microcosm and Hare-Lynx datasets are both open source and available from https://github.com/WillemBonnaffe/NODEBNGM.

## C.4 Hare-Lynx

This describes a real-world dataset of hare and lynx populations, replicating predator-prey dynamics [67]. We use the dataset made available by [79]. The dataset consists of a single trajectory of 92 time steps, and we use a train, val, test split of 70%, 15% and 15%, ensuring the splits are along the time dimension to maintain the integrity of temporal causality, following their chronological order. We further outline this dataset's system description and variable descriptions with the following prompt template as given in Appendix E.4.

# D  Benchmark Method Implementation Details

To be competitive we compare against popular black-box models, which, when modeling the dynamics of a system over time, becomes a form of ODE model, that is a neural ODE [14] with action inputs (**DyNODE**) [68]. We also compare against transparent dynamical equations derived from equation discovery methods for ODEs such as Sparse Identification of Nonlinear Dynamics (**SINDy**) [11]. Moreover, we compare against the ablations of our method, of the zero-shot generated HDTwin (**ZeroShot**) and this model with subsequently optimized parameters (**ZeroOptim**).

## DyNODE

DyNODE is a black-box neural network-based dynamics model [68], that models the underlying dynamics of a system by incorporating control into the standard neural ordinary differential equation framework [14]. We use a DyNODE model with 3-layer Multilayer Perceptron (MLP), with a hidden dimension of 128 units, with tanh activation functions, and make it competitive by using Xavier weight initialization [80]. To be competitive we use the same objective, optimizer and same hyperparameters for the optimizer that we use in HDTwinGen. That of an Adam optimizer [32], with a learning rate of 0.01, with a batch size of 1,000 and early stopping with a patience of 20, and train it for 2,000 epochs to ensure it converges.

## Causal Transformer

Causal Transformer is a state-of-the-art transformer model for estimating counterfactual outcomes [64]. Due to the complexity of the Causal Transformer, incorporating three separate transformer networks, each one for processing covariates, past treatments, and past outcomes, respectively—which is unique to estimating counterfactual outcomes in treatment effect settings; we implemented only a single transformer to model the past outcomes, which is applicable to our datasets and task domains. Specifically, this consists of a standard transformer encoder, where the input dataset is normalized to the training dataset. We encode input observed dimension of the state-action into an embedding vector dimension of size 250 through a linear layer, followed by the addition of a standard positional encoder [64]; this is then fed into a transformer encoder layer, with a head size of 10, dropout 0.1, and the output of this is then fed into a linear layer to reconstruct the next step ahead state, of size of the state dimension. We train this model using the AdamW [32] optimizer with a learning rate of 0.00005 and a step learning rate scheduler of step size 1.0 and gamma 0.95; we also implement gradient clipping to 0.7, with a batch size of 1,000 and early stopping with a patience of 20, and train it for 2,000 epochs to ensure it converges.

## RNN

Recurrent Neural Network [81] is a standard baseline that is widely used in autoregressive time series next step ahead prediction. We implement this where the input dataset is normalized to the training dataset. It consists of a gated recurrent unit RNN taking the state-action dimension in mapping it to a hidden dimension of size 250, with two layers. The output is then fed to a linear layer to convert the hidden dimension back to the state dimension to predict the next step ahead. To be competitive we use the same objective, optimizer and same hyperparameters for the optimizer that we use in HDTwinGen. That of an Adam optimizer [32], with a learning rate of 0.01, with a batch size of 1,000 and early stopping with a patience of 20, and train it for 2,000 epochs to ensure it converges.

## SINDy

Sparse Identification of Nonlinear Dynamics (SINDy) [11], is a data-driven framework that aims to discover the governing dynamical system equations directly from time-series data, discovering a white-box closed-form mathematical model. The algorithm works by iteratively performing sparse

regression on a library of candidate functions to identify the sparsest yet most accurate representation of the dynamical system.

In our implementation, we use a polynomial library of order two, which is a feature library of $\mathcal{L} = \{1, x_0, x_1, x_0 x_1\}$. Finite difference approximations are used to compute time derivatives from the input time-series data, of order one. Here the alpha parameter is kept constant at 0.5 across all experiments, and the sparsity threshold is set to 0.02 for all experiments, apart from the COVID-19 dataset where it is set to $1 \times 10^{-5}$.

**APHYNITY**

APHYNITY [60] is implemented using domain-specific expert models as defined in Appendix H.8 combined with a 3-layer MLP, with the same hyper-parameters as in [60].

**GP**

Genetic programming (GP) is implemented using the implementation and hyper-parameters from the baseline in [82].

**HDTwinGen**

See the section Appendix E for the implementation details. Specifically, **ZeroShot** and **ZeroOptim** are ablations of our method using the exact same setup, hyperparameters and prompts. Here ZeroShot generates one HDTwin, and does not fit its parameters, thus evaluating the loss of the model output directly from the LLM. Whereas ZeroOptim, repeats ZeroShot with the additional step of optimizing the parameters of the HDTwin that was generated—again using the same training as detailed in Appendix E.2.

# E    HDTwinGen Implementation Details

Our proposed method follows the framework as described in Section 4. We present pseudocode in Appendix E.1, how the code-generated HDTwins are trained in Appendix E.2, prompt templates in Appendix E.3, system requirements prompts in Appendix E.4 for each dataset, and we provide examples of training runs in Appendix J. Specifically, we find a top-K, where $K = 16$ is sufficient. Additionally, we use the LLM of GPT4-1106-Preview, with a temperature of 0.7.

## E.1    HDTwin pseudocode

---
**Algorithm 1** Pseudocode for Hybrid Digital Twin Generator Framework

---
1: **Input:** modeling context $\mathcal{S}^{context}$; training dataset $\mathcal{D}_{\text{train}}$, validation dataset $\mathcal{D}_{\text{val}}$, maximum generations $G$, top $K$ programs to consider, $\mathcal{R}$
2: **Output:** Best fitting hybrid model $f_{\theta, \omega(\theta)^*}$.
3: $\mathcal{P} \leftarrow \emptyset, H \leftarrow \emptyset$
4: **for** $g = 1$ **to** $G$ **do**
5:     $f_{\theta, \omega(\theta)} \sim LLM_{model}(H, \mathcal{P}^{(g)}, \mathcal{S}^{context})$ {Generate HDTwin from modeling agent}
6:     $\omega(\theta)^* = \arg\min_{\omega(\theta) \in \Omega(\theta)} \mathcal{L}(f_{\theta, \omega(\theta)}, \mathcal{D}_{\text{train}})$ {Fit the model}
7:     Compute validation loss per component and overall $\delta, \upsilon$
8:     $\mathcal{P}^{(g+1)} \leftarrow \mathcal{P}^{(g)} \oplus (f_{\theta, \omega(\theta)^*}, \delta, \upsilon)$ {Add HDTwin to the set of top-K HDTwins}
9:     $H \sim LLM_{eval}(\mathcal{R}, \mathcal{P}^{(g)})$ {Generate self-reflection on how to improve the HDTwin.}
10: **end for**
11: **Return:** $f_{\theta, \omega(\theta)^*}$ {The best fitting model that scored the lowest validation loss}

---

## E.2    Training HDTwins

Once the modeling agent has generated an HDTwin $f_{\theta, \omega(\theta)}$, it generates it as code. Specifically, it outputs code for a PyTorch [31] neural network module, where this code string is executed, and the module is then trained on the training dataset. The agent importantly observes a code skeleton within its system requirements context $\mathcal{S}^{context}$, of which examples of such a skeleton are given in Appendix E.4. However we stipulate that the skeleton must be a "torch.nn.Module", be called

"StateDifferential", and the parameters must be initialized, and it must define a forward function for computing the state differential of the state, where the state and action for that system are input variables to the function. The LLM is instructed to not to modify the code skeleton, only complete it, and return it. This makes it straightforward to process this from text, execute the module, and then train the model.

Specifically, we train the model on the training dataset, using the standard MSE loss Equation (5), optimizing using the Adam optimizer [32]. We use the same optimizer hyperparameters as the black-box neural network method, that of a learning rate of 0.01, with a batch size of 1,000 and early stopping with a patience of 20, and train it for 2,000 epochs to ensure it converges, to ensure fair comparison.

Once the model is trained, we compute the val MSE and val MSE per component, which corresponds to the val loss per state output dimension Equation (6). Notably, when we append the trained HDTwin back into $\mathcal{P}_g$ we include a string representation of it, which includes the values of any named parameters that were initialized in the model. We observe that feeding in the previous optimized parameters helps the LLM in subsequent generations to suggest good starting initial values for the named parameters, from which they can be further refined with the optimization step.

### E.3    HDTwinGen Prompt Templates

In the following we detail the prompt templates used. We always use the system prompt when interacting with the LLM.

#### System prompt

```
Objective: Write code to create an effective differential equation simulator for a given task.
Please note that the code should be fully functional. No placeholders.

You must act autonomously and you will receive no human input at any stage. You have to return as output the complete code for
    completing this task, and correctly improve the code to create the most accurate and realistic simulator possible.
You always write out the code contents. You always indent code with tabs.
You cannot visualize any graphical output. You exist within a machine. The code can include black box multi-layer perceptions where
    required.

Use the functions provided. When calling functions only provide a RFC8259 compliant JSON request following this format without
    deviation.
```

#### Defined function schema prompt

```
{
    "name": "complete_StateDifferential_code",
    "description": "Write out the code body for the 'StateDifferential' torch model.",
    "parameters": {
        "type": "object",
        "properties": {
            "StateDifferential_code": {
                "type": "string",
                "description": 'Code for the 'StateDifferential' torch model, inclusive of the model definition. If you are unsure,
                    take your best guess. This must be a nonempty string.',
            },
            "code_description": {
                "type": "string",
                "description": 'A concise description of the code model, indicating if it is a white box only or white and black box
                    model.',
            }
        },
        "required": ["StateDifferential_code", "code_description"],
    },
},
```

#### Modeling agent first task prompt

```
"""
You will get a system description to code a differential equation simulator for.

System Description:'''
{system_description}
'''

Modelling goals:'''
* The parameters of the model will be optimized to an observed training dataset with the given simulator.
* The observed training dataset has very few samples, and the model must be able to generalize to unseen data.
'''

Requirement Specification:'''
* The code generated should achieve the lowest possible validation loss, of 1e-10 or less.
* The code generated should be interpretable, and fit the dataset as accurately as possible.
'''

Skeleton code to fill in:'''
{skeleton_code}
'''

Useful to know:'''
* You are a code evolving machine, and you will be called {generations} times to generate code, and improve the code to achieve the
    lowest possible validation loss.
```

```
* The model defines the state differential and will be used with an ODE solver to fit the observed training dataset.
* You can use any parameters you want and any black box neural network components (multi-layer perceptrons); however, you have to
    define these.
* It is preferable to decompose the system into differential equations (compartments) if possible.
* You can use any unary functions, for example log, exp, power etc.
* Under no circumstance can you change the skeleton code function definitions, only fill in the code.
* The input tensors are vectors of shape (batch_size).
* Use initially white box models first and then switch to hybrid white and black box models for the residuals, only after no further
    best program iteration improvement with white box models.
* Make sure your code follows the exact code skeleton specification.
* Use PyTorch.
```

Think step-by-step, and then give the complete full working code. You are generating code for iteration {current_iteration} out of {
    generations}.
"""
```

## Reflection prompt

```
"""
You generated the following code completions, which then had their parameters optimized to the training dataset. Please reflect on
    how you can improve the code to minimize the validation loss to 1e-6 or less. The code examples are delineated by ###.

Here are your previous iterations the best programs generated. Use it to see if you have exhausted white box models, i.e. when a
    white box model repeats with the same val loss and then only add black box models to the white box models:```
{history_best_completions_str}
```

Here are the top code completions so far that you have generated, sorted for the lowest validation loss last:```
{completions}
```

Please reflect on how you can improve the code to fit the dataset as accurately as possible, and be interpretable. Think step-by-
    step. Provide only actionable feedback, that has direct changes to the code. Do not write out the code, only describe how it
    can be improved. Where applicable use the values of the optimized parameters to reason how the code can be improved to fit
    the dataset as accurately as possible. This is for generating new code for the next iteration {iteration} out of {self.config
    .run.generations}.
"""
```

## Modeling agent in subsequent generations

```
"""
Please now regenerate the code function, with the aim to improve the code to achieve a lower validation error. Use the feedback
    where applicable. You are generating code for iteration {generation_id} out of {self.config.run.generations} total iterations
    . When generating code if you are unsure about something, take your best guess. You have to generate code, and cannot give an
    empty string answer.

Please always only fill in the following code skeleton:```
{prompts.get_skeleton_code(self.env.env_name)}
```
You cannot change the code skeleton, or input variables.
"""
```

## E.4 HDTwinGen System Requirements Prompts

By following our proposed system requirements format, we constructed prompts for each of the datasets that we evaluated against, which are listed in the following.

### Lung Cancer (with Chemo. & Radio.)

```
You will get a system description to code a differential equation simulator for.

System Description:```
Prediction of Treatment Response for Combined Chemo and Radiation Therapy for Non-Small Cell Lung Cancer Patients Using a Bio-
    Mathematical Model

Here you must model the state differential of tumor_volume, and chemotherapy_drug_concentration; with the input actions of
    chemotherapy_dosage, and radiotherapy_dosage.

Description of the variables:
* tumor_volume: Volume of the tumor with units cm^3
* chemotherapy_drug_concentration: Concentration of the chemotherapy drug vinblastine with units mg/m^3
* chemotherapy_dosage: Dosage of the chemotherapy drug vinblastine with units mg/m^3
* radiotherapy_dosage: Dosage of the radiotherapy with units Gy

The time units is in days.

Additionally these variables have the ranges of:
* tumor_volume: [0.01433, 1170.861]
* chemotherapy_drug_concentration: [0, 9.9975]
* chemotherapy_dosage: [0, 5.0]
* radiotherapy_dosage: [0, 2.0]

The training dataset consists of 1000 patients, where each patient is observed for 60 days.
```

Modelling goals:```
* The parameters of the model will be optimized to an observed training dataset with the given simulator.
* The observed training dataset has very few samples, and the model must be able to generalize to unseen data.
```

Requirement Specification:```
* The code generated should achieve the lowest possible validation loss, of 1e-6 or less.
* The code generated should be interpretable, and fit the dataset as accurately as possible.
```

Skeleton code to fill in:```
class StateDifferential(nn.Module):
```

```
        def __init__(self):
            super(StateDifferential, self).__init__()
            # TODO: Fill in the code here

        def forward(self, tumor_volume: torch.Tensor, chemotherapy_drug_concentration: torch.Tensor, chemotherapy_dosage: torch.Tensor,
                radiotherapy_dosage: torch.Tensor) -> Tuple[torch.Tensor, torch.Tensor]:
            # TODO: Fill in the code here
            return (d_tumor_volume__dt, d_chemotherapy_drug_concentration__dt)
```

Useful to know:```
* You are a code evolving machine, and you will be called 20 times to generate code, and improve the code to achieve the lowest
    possible validation loss.
* The model defines the state differential and will be used with an ODE solver to fit the observed training dataset.
* You can use any parameters you want and any black box neural network components (multi-layer perceptrons); however, you have to
    define these.
* It is preferable to decompose the system into differential equations (compartments) if possible.
* You can use any unary functions, for example log, exp, power etc.
* Under no circumstance can you change the skeleton code function definitions, only fill in the code.
* The input tensors are vectors of shape (batch_size).
* Use initially white box models first and then switch to hybrid white and black box models for the residuals, only after no further
    best program iteration improvement with white box models.
* Make sure your code follows the exact code skeleton specification.
* Use PyTorch.
```

Think step-by-step, and then give the complete full working code. You are generating code for iteration 0 out of 20.

## Lung Cancer (with Chemo.)

You will get a system description to code a differential equation simulator for.

System Description:```
Prediction of Treatment Response for Combined Chemo and Radiation Therapy for Non-Small Cell Lung Cancer Patients Using a Bio-
    Mathematical Model

Here you must model the state differential of tumor_volume, and chemotherapy_drug_concentration; with the input actions of
    chemotherapy_dosage.

Description of the variables:
* tumor_volume: Volume of the tumor with units cm^3
* chemotherapy_drug_concentration: Concentration of the chemotherapy drug vinblastine with units mg/m^3
* chemotherapy_dosage: Dosage of the chemotherapy drug vinblastine with units mg/m^3

The time units is in days.

Additionally these variables have the ranges of:
* tumor_volume: [0.64196031, 1260.60290569]
* chemotherapy_drug_concentration: [0, 9.9975]
* chemotherapy_dosage: [0, 5.0]

The training dataset consists of 1000 patients, where each patient is observed for 60 days.
```

Modelling goals:```
* The parameters of the model will be optimized to an observed training dataset with the given simulator.
* The observed training dataset has very few samples, and the model must be able to generalize to unseen data.
```

Requirement Specification:```
* The code generated should achieve the lowest possible validation loss, of 1e-6 or less.
* The code generated should be interpretable, and fit the dataset as accurately as possible.
```

Skeleton code to fill in:```
class StateDifferential(nn.Module):
    def __init__(self):
        super(StateDifferential, self).__init__()
        # TODO: Fill in the code here

    def forward(self, tumor_volume: torch.Tensor, chemotherapy_drug_concentration: torch.Tensor, chemotherapy_dosage: torch.Tensor)
            -> Tuple[torch.Tensor, torch.Tensor]:
        # TODO: Fill in the code here
        return (d_tumor_volume__dt, d_chemotherapy_drug_concentration__dt)
```

Useful to know:```
* You are a code evolving machine, and you will be called 20 times to generate code, and improve the code to achieve the lowest
    possible validation loss.
* The model defines the state differential and will be used with an ODE solver to fit the observed training dataset.
* You can use any parameters you want and any black box neural network components (multi-layer perceptrons); however, you have to
    define these.
* It is preferable to decompose the system into differential equations (compartments) if possible.
* You can use any unary functions, for example log, exp, power etc.
* Under no circumstance can you change the skeleton code function definitions, only fill in the code.
* The input tensors are vectors of shape (batch_size).
* Use initially white box models first and then switch to hybrid white and black box models for the residuals, only after no further
    best program iteration improvement with white box models.
* Make sure your code follows the exact code skeleton specification.
* Use PyTorch.
```

Think step-by-step, and then give the complete full working code. You are generating code for iteration 0 out of 20.

## Lung Cancer

You will get a system description to code a differential equation simulator for.

System Description:```
Prediction of Treatment Response for Combined Chemo and Radiation Therapy for Non-Small Cell Lung Cancer Patients Using a Bio-
    Mathematical Model

Here you must model the state differential of tumor_volume. There are not treatments applied.

Description of the variables:
* tumor_volume: Volume of the tumor with units cm^3

The time units is in days.

Additionally these variables have the ranges of:
* tumor_volume: [0.64196031, 4852.45734281]

The training dataset consists of 1000 patients, where each patient is observed for 60 days.
```

Modelling goals:```
* The parameters of the model will be optimized to an observed training dataset with the given simulator.
* The observed training dataset has very few samples, and the model must be able to generalize to unseen data.
```

Requirement Specification:```
* The code generated should achieve the lowest possible validation loss, of 1e-6 or less.
* The code generated should be interpretable, and fit the dataset as accurately as possible.
```

Skeleton code to fill in:```
class StateDifferential(nn.Module):
    def __init__(self):
        super(StateDifferential, self).__init__()
        # TODO: Fill in the code here

    def forward(self, tumor_volume: torch.Tensor) -> Tuple[torch.Tensor]:
        # TODO: Fill in the code here
        return (d_tumor_volume__dt)
```

Useful to know:```
* You are a code evolving machine, and you will be called 20 times to generate code, and improve the code to achieve the lowest possible validation loss.
* The model defines the state differential and will be used with an ODE solver to fit the observed training dataset.
* You can use any parameters you want and any black box neural network components (multi-layer perceptrons); however, you have to define these.
* It is preferable to decompose the system into differential equations (compartments) if possible.
* You can use any unary functions, for example log, exp, power etc.
* Under no circumstance can you change the skeleton code function definitions, only fill in the code.
* The input tensors are vectors of shape (batch_size).
* Use initially white box models first and then switch to hybrid white and black box models for the residuals, only after no further best program iteration improvement with white box models.
* Make sure your code follows the exact code skeleton specification.
* Use PyTorch.
```

Think step-by-step, and then give the complete full working code. You are generating code for iteration 0 out of 20.

# Hare-Lynx

You will get a system description to code a differential equation simulator for.

System Description:```
"Modeling Di-Trophic Prey-Predator Dynamics in a Hare and Lynx Ecological System

Here you must model the state differential of hare_population, and lynx_population; with the additional input of time_in_years. This aims to simulate the population dynamics within a simplified di-trophic ecological system comprising prey (hares), and predators (lynxes). The interactions include direct predation and competition for resources, mirroring natural predator-prey mechanisms.

Description of the variables:
* hare_population: Annual count of hare pelts, serving as a proxy for the hare population size, in tens of thousands.
* lynx_population: Annual count of lynx pelts, serving as a proxy for the lynx population size, in tens of thousands.

The model should capture the dynamics of these populations, reflecting the di-trophic prey-predator interactions, and predict the population sizes based on historical data. The data exhibits 10-year long characteristic oscillations due to prey-predator dynamics.

Additionally these variables have the ranges of:
* hare_population: [1.80, 152.65]
* lynx_population: [3.19, 79.35]
* time_in_years: [1845, 1935]

The training dataset consists of 63 time steps, validation and training dataset consists of 14 time steps each.

```

Modelling goals:```
* The parameters of the model will be optimized to an observed training dataset with the given simulator.
* The observed training dataset has very few samples, and the model must be able to generalize to unseen data.
```

Requirement Specification:```
* The code generated should achieve the lowest possible validation loss, of 1e-6 or less.
* The code generated should be interpretable, and fit the dataset as accurately as possible.
```

Skeleton code to fill in:```
class StateDifferential(nn.Module):
    def __init__(self):
        super(StateDifferential, self).__init__()
        # TODO: Fill in the code here

    def forward(self, hare_population: torch.Tensor, lynx_population: torch.Tensor, time_in_years: torch.Tensor) -> Tuple[torch.Tensor, torch.Tensor]:
        # TODO: Fill in the code here
        return (d_hare_population__dt, d_lynx_population__dt)
```

Useful to know:```
* You are a code evolving machine, and you will be called 20 times to generate code, and improve the code to achieve the lowest possible validation loss.
* The model defines the state differential and will be used with an ODE solver to fit the observed training dataset.

* You can use any parameters you want and any black box neural network components (multi-layer perceptrons); however, you have to define these.
* It is preferable to decompose the system into differential equations (compartments) if possible.
* You can use any unary functions, for example log, exp, power etc.
* Under no circumstance can you change the skeleton code function definitions, only fill in the code.
* The input tensors are vectors of shape (batch_size).
* Use initially white box models first and then switch to hybrid white and black box models for the residuals, only after no further best program iteration improvement with white box models.
* Make sure your code follows the exact code skeleton specification.
* Use PyTorch.
```

Think step-by-step, and then give the complete full working code. You are generating code for iteration 0 out of 20.

## Plankton Microcosm

You will get a system description to code a differential equation simulator for.

System Description:```
"Modeling Artificial Tri-Trophic Prey-Predator Oscillations in a Simplified Ecological System

Here you must model the state differential of algae_population, flagellate_population, and rotifer_population; with no input actions. This aims to simulate the population dynamics within a simplified tri-trophic ecological system comprising prey (algae), intermediate predators (flagellates), and top predators (rotifers). The interactions include direct predation and competition for resources, mirroring natural intraguild predation mechanisms.

Description of the variables:
* prey_population: Total count of algae, serving as the primary prey
* intermediate_population: Total count of flagellates, acting as intermediate predators and prey
* top_predators_population: Total count of rotifers, representing top predators

The dataset encapsulates daily population counts across multiple simulated ecosystems over a period of 100 days, allowing for the analysis of temporal oscillations and phase lags between species.

Additionally these variables have the ranges of:
* prey_population: [0.095898, 2.469735]
* intermediate_population: [0.008438, 1.500000]
* top_predators_population: [0.030316, 0.739244]

The training dataset consists of 70 time steps, validation and training dataset consists of 15 time steps each.

```

Modelling goals:```
* The parameters of the model will be optimized to an observed training dataset with the given simulator.
* The observed training dataset has very few samples, and the model must be able to generalize to unseen data.
```

Requirement Specification:```
* The code generated should achieve the lowest possible validation loss, of 1e-6 or less.
* The code generated should be interpretable, and fit the dataset as accurately as possible.
```

Skeleton code to fill in:```
class StateDifferential(nn.Module):
    def __init__(self):
        super(StateDifferential, self).__init__()
        # TODO: Fill in the code here

    def forward(self, prey_population: torch.Tensor, intermediate_population: torch.Tensor, top_predators_population: torch.Tensor)
        -> Tuple[torch.Tensor, torch.Tensor, torch.Tensor]:
        # TODO: Fill in the code here
        return (d_prey_population__dt, d_intermediate_population__dt, d_top_predators_population__dt)
```

Useful to know:```
* You are a code evolving machine, and you will be called 20 times to generate code, and improve the code to achieve the lowest possible validation loss.
* The model defines the state differential and will be used with an ODE solver to fit the observed training dataset.
* You can use any parameters you want and any black box neural network components (multi-layer perceptrons); however, you have to define these.
* It is preferable to decompose the system into differential equations (compartments) if possible.
* You can use any unary functions, for example log, exp, power etc.
* Under no circumstance can you change the skeleton code function definitions, only fill in the code.
* The input tensors are vectors of shape (batch_size).
* Use initially white box models first and then switch to hybrid white and black box models for the residuals, only after no further best program iteration improvement with white box models.
* Make sure your code follows the exact code skeleton specification.
* Use PyTorch.
```

Think step-by-step, and then give the complete full working code. You are generating code for iteration 0 out of 20.

## COVID-19

You will get a system description to code a differential equation simulator for.

System Description:```
Prediction model of COVID-19 Epidemic Dynamics

Here you must model the state differential of susceptible, exposed, infected and recovered; with the input action of a constant total_population. There are no interventions applied. Here the states are normalized ratios of the total fixed population.

Description of the variables:
* susceptible: Ratio of the population that is susceptible to the virus.
* exposed: Ratio of the population that is exposed to the virus, not yet infectious.
* infected: Ratio of the population that is actively carrying and transmitting the virus.
* recovered: Ratio of the population that have recovered from the virus, including those who are deceased.
* total_population: Total population of the country, a constant.

The time units is in days.

Additionally these variables have the ranges of:
* susceptible: [0, 1]

```
* exposed: [0, 1]
* infected: [0, 1]
* recovered: [0, 1]
* total_population: [10000, 10000]

The training dataset consists of 24 countries, where each country is observed for 60 days.
```

Modelling goals:```
* The parameters of the model will be optimized to an observed training dataset with the given simulator.
* The observed training dataset has very few samples, and the model must be able to generalize to unseen data.
```

Requirement Specification:```
* The code generated should achieve the lowest possible validation loss, of 1e-10 or less.
* The code generated should be interpretable, and fit the dataset as accurately as possible.
```

Skeleton code to fill in:```
class StateDifferential(nn.Module):
    def __init__(self):
        super(StateDifferential, self).__init__()
        # TODO: Fill in the code here

    def forward(self, susceptible: torch.Tensor, exposed: torch.Tensor, infected: torch.Tensor, recovered: torch.Tensor,
        total_population: torch.Tensor) -> Tuple[torch.Tensor, torch.Tensor, torch.Tensor, torch.Tensor]:
        # TODO: Fill in the code here
        return (d_susceptible__dt, d_exposed__dt, d_infected__dt, d_recovered__dt)
```

Useful to know:```
* You are a code evolving machine, and you will be called 20 times to generate code, and improve the code to achieve the lowest
    possible validation loss.
* The model defines the state differential and will be used with an ODE solver to fit the observed training dataset.
* You can use any parameters you want and any black box neural network components (multi-layer perceptrons); however, you have to
    define these.
* It is preferable to decompose the system into differential equations (compartments) if possible.
* You can use any unary functions, for example log, exp, power etc.
* Under no circumstance can you change the skeleton code function definitions, only fill in the code.
* The input tensors are vectors of shape (batch_size).
* Use initially white box models first and then switch to hybrid white and black box models for the residuals, only after no further
    best program iteration improvement with white box models.
* Make sure your code follows the exact code skeleton specification.
* Use PyTorch.
```

Think step-by-step, and then give the complete full working code. You are generating code for iteration 0 out of 20.

# F    Model Optimization Losses

We consider the optimization loss of mean squared error on a dataset $\mathcal{D}$ and also consider a higher fidelity mean squared error loss per component.

**MSE Loss.** Specifically, we optimize the following mean squared error objective,

$$\mathcal{L}(\theta, \mathcal{D}) = \frac{1}{N \times T} \sum_{n=1}^{N} \sum_{i=0}^{T^n} ||f_{\theta,\omega(\theta)}(x^{(n)}(t_i), u^{(n)}(t_i), t_i)\Delta t - y^{(n)}(t_i)||^2 \tag{5}$$

where $N \times T$ is the total number of state-action pairs in the dataset. For a given model find the parameters $\theta^*$ that minimize this loss, i.e. $\theta^* = \arg\min_\theta \mathcal{L}(\theta, \mathcal{D}_{\text{train}})$. Here we optimize $\theta$ by stochastic gradient descent, using the Adam optimizer [32], however, we note other optimization algorithms could also be used such as black box optimizers.

**MSE Loss per component.** We seek to collect detailed quantitative statistics on how well the generated trained system model performs. Therefore, we collect the validation loss per component. Here, we use $^{(j)}$ to indicate the predictions for the $j^{th}$ component.

$$\omega_m(\theta^*, \mathcal{D}_{\text{val}}) = \frac{1}{N_{\text{val}} \times T} \sum_{n=1}^{N_{\text{val}}} \sum_{i=0}^{T^n_{\text{val}}} \left(f^{(j)}_{\theta,\omega(\theta)}(x^{(n)}(t_i), u^{(n)}(t_i), t_i)\Delta t - y^{(n)}_j(t_i)\right)^2 \tag{6}$$

and collect these scalar validation losses per component into a vector $\omega = [\omega_1, \omega_2, \ldots, \omega_m]$, and compute its mean as $v = \frac{1}{m} \sum_{j=1}^{m} \omega_j(\theta^*, \mathcal{D}_{\text{val}})$ i.e. the validation loss.

# G    Evaluation Metrics

We employ mean squared error (MSE) to evaluate the benchmark methods on a held-out test dataset of state-action trajectories, denoted as $\mathcal{D}_{\text{test}}$, using the loss defined in Equation (5) and report this as $\mathcal{T}_{MSE}$. Each metric is averaged over ten runs with different random seeds, and we present these

averages along with their 95% mean confidence intervals[7]. For each random seed run, we generate a new train, validation and test dataset independently, when we have access to a simulator. Additionally, when sampling a dataset from a simulator, we collect datasets of the same number of trajectories for the validation and test set as was used to generate the training set. We then train each baseline on the training dataset and use the validation dataset for early stopping when the method supports this. We then evaluate the performance of each baseline on the test dataset. We repeat this process for each random seed run. We perform all experiments and training using a single Intel Core i9-12900K CPU @ 3.20GHz, 64GB RAM with an Nvidia RTX3090 GPU 24GB.

# H  Additional Results

## H.1  Out-of-distribution Experiment and Setup

To explore the out-of-distribution shifts, we adapt the Lung Cancer (with Chemo. & Radio.) simulator to have a training dataset of tumor volumes in a range that is outside of the tumor volume range within the test set over all trajectories. To do this we sampled a training dataset, with starting state tumor volumes sampled from $x \sim \mathcal{U}(0, 574)$ and then collected $1,000$ trajectories for 60 time steps. We then sampled a test dataset out of distribution from that seen in training, with a starting initial state of $x \sim \mathcal{U}(804, 1149)$. To ensure that the test state-action states are completely out of the training distribution we slowed down the time interval of the simulator to that of per hour, rather than day, i.e., simulating at $\Delta t = \frac{1}{24}$ rather than $\Delta t = 1$ day resolution. We then also verified that the range of cancer volumes seen within the test set do not overlap at all with those in the training set, especially throughout and at the end of the trajectory.

## H.2  COVID-19 Unobserved Intervention Experiment and Setup

To assess whether HDTwinGen can understand and modify its HDTwin, we setup an experiment to see if the modeling agent can adapt an already optimized and best-discovered HDTwin for an unobserved intervention, that is a change of the true system state function dynamics, that are latent.

We constructed a scenario with our COVID-19 simulator, to have a realistic intervention of a lockdown policy, which reduces the physical distance of individuals to one another. In COVASIM we implemented this, where this happens at day 19, dramatically altering how the COVASIM simulator behaves after the intervention is applied. This is known to reduce the *effective contact rate $\beta$* parameter in an SEIR model [83], for the intervention this approximately corresponds to the $\beta$ parameter reducing its value by 75% after the lockdown intervention is applied. Importantly we now sampled a training dataset, and validation dataset from the simulator for before the intervention was applied, and only sampled a test dataset after the intervention was applied.

We took the best-found trained HDTwin model for COVID-19, itself created a HDTwin model that is an SEIR model with black-box residual components, as seen in Appendix I. The agent then was specifically instructed to attempt to adapt this HDTwin model to this unobserved intervention, that was purely described in words. The LLM was able to reason about it, and similarly decreased the HDTwin models internal $\beta$ parameter by approximately 70%, leading to un-observed modelling of such an intervention, when rolling out from the current state, from day 19. This shows the utility of such a hybrid DT framework. Interestingly, the competing methods, such as the black-box method, DyNODE and SINDy incorrectly continue the expected trajectory as they are unaware that the underlying system has been intervened on, and its behavior is now different from what they have learnt to optimize to.

## H.3  HDTwinGen top-1 decreases over time

We observe in Figure 6, that averaged over 10 random seed runs, the top-K, specifically the top-K HDTwin found at each iteration step decreases, implying on average the HDTwin improves in the iteration loop in the beginning stages.

---

[7]We use the code at https://stackoverflow.com/questions/15033511/compute-a-confidence-interval-from-sample-data to compute these

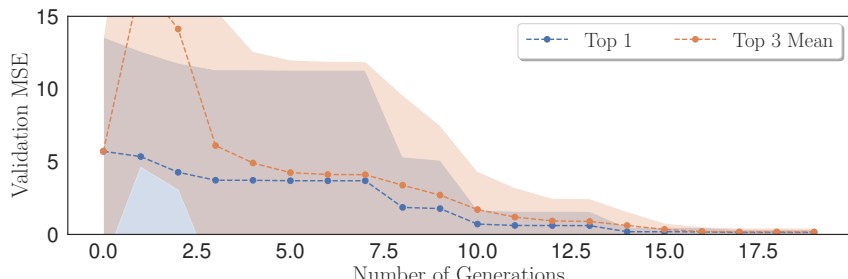

Figure 6: **Average evolution over generations for HDTwinGen**—for the Lung Cancer (with Chemo. & Radio.)

### H.4  HDTwinGen Evolution

Here we list out the specific HDTwins that have been generated at generation steps and are labeled in Figure 3.

Iteration 1. Val Loss: 4.414876937866211 Description: White box only.

```
Val Loss: 4.41 (Where the val loss per dimension is tumor_volume val loss: 2.77, chemotherapy_drug_concentration val loss: 6.06)
        Iteration: 1
###
```
import torch
import torch.nn as nn
from typing import Tuple

class StateDifferential(nn.Module):
    def __init__(self):
        super(StateDifferential, self).__init__()
        # Parameters for the tumor growth model
        self.alpha = nn.Parameter(torch.tensor(0.1))   # Tumor growth rate
        self.beta = nn.Parameter(torch.tensor(0.05)) # Tumor death rate due to chemotherapy
        self.gamma = nn.Parameter(torch.tensor(0.03)) # Tumor death rate due to radiotherapy

    def forward(self, tumor_volume: torch.Tensor, chemotherapy_drug_concentration: torch.Tensor, chemotherapy_dosage: torch.Tensor,
            radiotherapy_dosage: torch.Tensor) -> Tuple[torch.Tensor, torch.Tensor]:
        # Tumor volume change rate
        d_tumor_volume__dt = self.alpha * tumor_volume - self.beta * chemotherapy_drug_concentration * tumor_volume - self.gamma *
            radiotherapy_dosage * tumor_volume

        # Chemotherapy drug concentration change rate
        d_chemotherapy_drug_concentration__dt = chemotherapy_dosage - chemotherapy_drug_concentration  # Assuming a simple model
            where the drug is administered and then decays

        return (d_tumor_volume__dt, d_chemotherapy_drug_concentration__dt)
```
optimized_parameters = {'alpha': 0.04550161585211754, 'beta': 0.02731170691549778, 'gamma': 0.0489218533039093}
###
```

Iteration 2. Val Loss: 4.233582019805908 Description: White box model with logistic growth for tumor volume and decay rate for chemotherapy drug concentration.

```
Val Loss: 4.23 (Where the val loss per dimension is tumor_volume val loss: 8.47, chemotherapy_drug_concentration val loss: 8.95e−14)
        Iteration: 2
###
```
import torch
import torch.nn as nn
from typing import Tuple

class StateDifferential(nn.Module):
    def __init__(self):
        super(StateDifferential, self).__init__()
        # Parameters for the tumor growth model
        self.alpha = nn.Parameter(torch.tensor(0.1))   # Tumor growth rate
        self.beta = nn.Parameter(torch.tensor(0.05)) # Tumor death rate due to chemotherapy
        self.gamma = nn.Parameter(torch.tensor(0.03)) # Tumor death rate due to radiotherapy
        self.kappa = nn.Parameter(torch.tensor(1000.0)) # Carrying capacity of the environment
        self.delta = nn.Parameter(torch.tensor(0.01)) # Decay rate of the chemotherapy drug

    def forward(self, tumor_volume: torch.Tensor, chemotherapy_drug_concentration: torch.Tensor, chemotherapy_dosage: torch.Tensor,
            radiotherapy_dosage: torch.Tensor) -> Tuple[torch.Tensor, torch.Tensor]:
        # Logistic growth model for tumor volume
        d_tumor_volume__dt = self.alpha * tumor_volume * (1 − tumor_volume / self.kappa) − self.beta *
            chemotherapy_drug_concentration * tumor_volume − self.gamma * radiotherapy_dosage * tumor_volume

        # Chemotherapy drug concentration change rate with decay
        d_chemotherapy_drug_concentration__dt = chemotherapy_dosage − self.delta * chemotherapy_drug_concentration

        return (d_tumor_volume__dt, d_chemotherapy_drug_concentration__dt)
```
optimized_parameters = {'alpha': 0.05583872273564339, 'beta': 0.026076778769493103, 'gamma': 0.037345148622989655, 'kappa':
    1015.9735107421875, 'delta': 0.5000000596046448}
###
```

## Iteration 3. Val Loss: 2.902040958404541 Description: White box model with logistic growth for tumor volume, interaction term for combined therapy effect, and bi-exponential decay for chemotherapy drug concentration.

```
Val Loss: 2.9 (Where the val loss per dimension is tumor_volume val loss: 5.36, chemotherapy_drug_concentration val loss: 0.448)
        Iteration: 3
###
```
```
import torch
import torch.nn as nn
from typing import Tuple

class StateDifferential(nn.Module):
    def __init__(self):
        super(StateDifferential, self).__init__()
        # Parameters for the tumor growth model
        self.alpha = nn.Parameter(torch.tensor(0.05583872273564339))  # Tumor growth rate
        self.beta = nn.Parameter(torch.tensor(0.026076778769493103)) # Tumor death rate due to chemotherapy
        self.gamma = nn.Parameter(torch.tensor(0.037345148622989655)) # Tumor death rate due to radiotherapy
        self.kappa = nn.Parameter(torch.tensor(1015.9735107421875)) # Carrying capacity of the environment
        self.delta = nn.Parameter(torch.tensor(0.5000000596046448)) # Decay rate of the chemotherapy drug
        self.eta = nn.Parameter(torch.tensor(0.01)) # Interaction term between chemotherapy and radiotherapy

    def forward(self, tumor_volume: torch.Tensor, chemotherapy_drug_concentration: torch.Tensor, chemotherapy_dosage: torch.Tensor,
            radiotherapy_dosage: torch.Tensor) -> Tuple[torch.Tensor, torch.Tensor]:
        # Logistic growth model for tumor volume with interaction term
        d_tumor_volume__dt = self.alpha * tumor_volume * (1 - tumor_volume / self.kappa) - self.beta *
                chemotherapy_drug_concentration * tumor_volume - self.gamma * radiotherapy_dosage * tumor_volume - self.eta *
                chemotherapy_drug_concentration * radiotherapy_dosage * tumor_volume

        # Bi-exponential decay model for chemotherapy drug concentration
        d_chemotherapy_drug_concentration__dt = chemotherapy_dosage - self.delta * chemotherapy_drug_concentration**2

        return (d_tumor_volume__dt, d_chemotherapy_drug_concentration__dt)
```
```
optimized_parameters = {'alpha': 0.0490952804684639, 'beta': 0.023840805515646935, 'gamma': 0.03170385956764221, 'kappa':
    1032.3638916015625, 'delta': 0.06566499918699265, 'eta': 0.002417867537587881}
###
```

## Iteration 4: Val loss: 1.2533262968063354 Description: White and black box model with logistic growth for tumor volume, Michaelis-Menten saturation for chemotherapy effect, sigmoid function for radiotherapy effect, resistance development, and a black box MLP for residuals.

```
Val Loss: 1.25 (Where the val loss per dimension is tumor_volume val loss: 2.49, chemotherapy_drug_concentration val loss: 0.0203)
        Iteration: 4
###
```
```
import torch
import torch.nn as nn
from typing import Tuple

class StateDifferential(nn.Module):
    def __init__(self):
        super(StateDifferential, self).__init__()
        # Parameters for the tumor growth model
        self.alpha = nn.Parameter(torch.tensor(0.0490952804684639))  # Tumor growth rate
        self.beta = nn.Parameter(torch.tensor(0.023840805515646935)) # Tumor death rate due to chemotherapy
        self.gamma = nn.Parameter(torch.tensor(0.03170385956764221)) # Tumor death rate due to radiotherapy
        self.kappa = nn.Parameter(torch.tensor(1032.3638916015625)) # Carrying capacity of the environment
        self.delta = nn.Parameter(torch.tensor(0.06566499918699265)) # Decay rate of the chemotherapy drug
        self.eta = nn.Parameter(torch.tensor(0.002417867537587881)) # Interaction term between chemotherapy and radiotherapy
        self.theta = nn.Parameter(torch.tensor(0.5)) # Michaelis-Menten saturation constant for chemotherapy
        self.rho = nn.Parameter(torch.tensor(0.5)) # Sigmoid steepness for radiotherapy effect
        self.zeta = nn.Parameter(torch.tensor(0.1)) # Resistance development rate

        # Black box component for capturing residuals
        self.residual_mlp = nn.Sequential(
            nn.Linear(4, 10),
            nn.ReLU(),
            nn.Linear(10, 2)
        )

    def forward(self, tumor_volume: torch.Tensor, chemotherapy_drug_concentration: torch.Tensor, chemotherapy_dosage: torch.Tensor,
            radiotherapy_dosage: torch.Tensor) -> Tuple[torch.Tensor, torch.Tensor]:
        # Logistic growth model for tumor volume with interaction term and resistance
        resistance = 1 + self.zeta * tumor_volume
        d_tumor_volume__dt = self.alpha * tumor_volume * (1 - tumor_volume / self.kappa) - (self.beta *
                chemotherapy_drug_concentration * tumor_volume) / (self.theta + chemotherapy_drug_concentration) - self.gamma *
                radiotherapy_dosage * tumor_volume / (1 + torch.exp(-self.rho * (radiotherapy_dosage - 1))) - self.eta *
                chemotherapy_drug_concentration * radiotherapy_dosage * tumor_volume / resistance

        # Bi-exponential decay model for chemotherapy drug concentration
        d_chemotherapy_drug_concentration__dt = chemotherapy_dosage - self.delta * chemotherapy_drug_concentration**2

        # Black box residual component
        residuals = self.residual_mlp(torch.stack((tumor_volume, chemotherapy_drug_concentration, chemotherapy_dosage,
                radiotherapy_dosage), dim=1))

        # Combine white box model with residuals
        d_tumor_volume__dt += residuals[:, 0]
        d_chemotherapy_drug_concentration__dt += residuals[:, 1]

        return (d_tumor_volume__dt, d_chemotherapy_drug_concentration__dt)
```
```
optimized_parameters = {'alpha': 0.029592309147119522, 'beta': 0.27041515707969666, 'gamma': 0.08552337437868118, 'kappa':
    1032.418701171875, 'delta': 0.03764118626713753, 'eta': 0.037968311458826065, 'theta': 5.688083648681641, 'rho':
    0.444132924079895, 'zeta': 0.1205417588353157}
###
```

## Iteration 5. Val loss: 0.1756206750869751 Description: White and black box model with dynamic carrying capacity and decay rate, enhanced interaction term, and an expanded black box MLP for residuals.

Val Loss: 0.176 (Where the val loss per dimension is tumor_volume val loss: 0.335, chemotherapy_drug_concentration val loss: 0.0163)
        Iteration: 5
###
```
import torch
import torch.nn as nn
from typing import Tuple

class StateDifferential(nn.Module):
    def __init__(self):
        super(StateDifferential, self).__init__()
        # Parameters for the tumor growth model
        self.alpha = nn.Parameter(torch.tensor(0.029592309147119522))  # Tumor growth rate
        self.beta = nn.Parameter(torch.tensor(0.27041515707969666)) # Tumor death rate due to chemotherapy
        self.gamma = nn.Parameter(torch.tensor(0.08552337437868118)) # Tumor death rate due to radiotherapy
        self.kappa_base = nn.Parameter(torch.tensor(1032.418701171875)) # Base carrying capacity of the environment
        self.kappa_mod = nn.Parameter(torch.tensor(0.1)) # Modifier for carrying capacity based on treatment
        self.delta_base = nn.Parameter(torch.tensor(0.03764118626713753)) # Base decay rate of the chemotherapy drug
        self.delta_mod = nn.Parameter(torch.tensor(0.1)) # Modifier for decay rate based on tumor volume
        self.eta = nn.Parameter(torch.tensor(0.037968311458826065)) # Interaction term between chemotherapy and radiotherapy
        self.theta = nn.Parameter(torch.tensor(5.688083648681641)) # Michaelis-Menten saturation constant for chemotherapy
        self.rho = nn.Parameter(torch.tensor(0.444132924079895)) # Sigmoid steepness for radiotherapy effect
        self.zeta = nn.Parameter(torch.tensor(0.1205417588353157)) # Resistance development rate

        # Black box component for capturing residuals
        self.residual_mlp = nn.Sequential(
            nn.Linear(4, 20),
            nn.ReLU(),
            nn.Linear(20, 20),
            nn.ReLU(),
            nn.Linear(20, 2)
        )

    def forward(self, tumor_volume: torch.Tensor, chemotherapy_drug_concentration: torch.Tensor, chemotherapy_dosage: torch.Tensor,
            radiotherapy_dosage: torch.Tensor) -> Tuple[torch.Tensor, torch.Tensor]:
        # Adjusted carrying capacity based on treatment
        kappa = self.kappa_base + self.kappa_mod * (chemotherapy_dosage + radiotherapy_dosage)

        # Adjusted decay rate based on tumor volume
        delta = self.delta_base + self.delta_mod * tumor_volume

        # Logistic growth model for tumor volume with interaction term and resistance
        resistance = 1 + self.zeta * tumor_volume
        d_tumor_volume__dt = self.alpha * tumor_volume * (1 - tumor_volume / kappa) - (self.beta * chemotherapy_drug_concentration *
                tumor_volume) / (self.theta + chemotherapy_drug_concentration) - self.gamma * radiotherapy_dosage * tumor_volume /
                (1 + torch.exp(-self.rho * (radiotherapy_dosage - 1))) - self.eta * chemotherapy_drug_concentration *
                radiotherapy_dosage * tumor_volume / resistance

        # Non-linear decay model for chemotherapy drug concentration
        d_chemotherapy_drug_concentration__dt = chemotherapy_dosage - delta * torch.pow(chemotherapy_drug_concentration, 1.5)

        # Black box residual component
        residuals = self.residual_mlp(torch.stack((tumor_volume, chemotherapy_drug_concentration, chemotherapy_dosage,
            radiotherapy_dosage), dim=1))

        # Combine white box model with residuals
        d_tumor_volume__dt += residuals[:, 0]
        d_chemotherapy_drug_concentration__dt += residuals[:, 1]

        return (d_tumor_volume__dt, d_chemotherapy_drug_concentration__dt)
```
optimized_parameters = {'alpha': 0.02282065525650978, 'beta': 0.343279629945755, 'gamma': 0.08097328990697861, 'kappa_base':
        1031.5604248046875, 'kappa_mod': -2.13154935836792, 'delta_base': -0.02879924327135086, 'delta_mod': 0.0003302362456452015,
        'eta': 0.0037569233682006598, 'theta': 8.673722267150879, 'rho': 0.37428149580955505, 'zeta': 0.156700998544693}
###

## Iteration 12. Val loss: 0.07228073477745056 Description: White and black box model with logistic growth for tumor volume, Michaelis-Menten saturation for chemotherapy effect, sigmoid function for radiotherapy effect, resistance development, and a simplified black box MLP for residuals.

Val Loss: 0.0723 (Where the val loss per dimension is tumor_volume val loss: 0.133, chemotherapy_drug_concentration val loss: 0.012)
        Iteration: 11
###
```
class StateDifferential(nn.Module):
    def __init__(self):
        super(StateDifferential, self).__init__()
        # Parameters for the tumor growth model
        self.alpha = nn.Parameter(torch.tensor(0.03))  # Tumor growth rate
        self.beta = nn.Parameter(torch.tensor(0.4)) # Tumor death rate due to chemotherapy
        self.gamma = nn.Parameter(torch.tensor(0.08)) # Tumor death rate due to radiotherapy
        self.kappa_base = nn.Parameter(torch.tensor(1030.0)) # Base carrying capacity of the environment
        self.kappa_mod = nn.Parameter(torch.tensor(-2.0)) # Modifier for carrying capacity based on treatment
        self.delta_base = nn.Parameter(torch.tensor(0.1)) # Base decay rate of the chemotherapy drug
        self.delta_mod = nn.Parameter(torch.tensor(0.0003)) # Modifier for decay rate based on tumor volume
        self.eta = nn.Parameter(torch.tensor(0.004)) # Interaction term between chemotherapy and radiotherapy
        self.theta = nn.Parameter(torch.tensor(10.0)) # Michaelis-Menten saturation constant for chemotherapy
        self.rho = nn.Parameter(torch.tensor(0.3)) # Sigmoid steepness for radiotherapy effect
        self.zeta = nn.Parameter(torch.tensor(0.15)) # Resistance development rate

        # Black box component for capturing residuals
        self.residual_mlp = nn.Sequential(
            nn.Linear(4, 16),
            nn.LeakyReLU(0.1),
            nn.Linear(16, 8),
            nn.LeakyReLU(0.1),
            nn.Linear(8, 2)
        )
```

```
def forward(self, tumor_volume: torch.Tensor, chemotherapy_drug_concentration: torch.Tensor, chemotherapy_dosage: torch.Tensor,
        radiotherapy_dosage: torch.Tensor) -> Tuple[torch.Tensor, torch.Tensor]:
    # Adjusted carrying capacity based on treatment
    kappa = self.kappa_base + self.kappa_mod * (chemotherapy_dosage + radiotherapy_dosage)

    # Adjusted decay rate based on tumor volume
    delta = self.delta_base + self.delta_mod * tumor_volume

    # Logistic growth model for tumor volume with interaction term and resistance
    resistance = 1 + self.zeta * tumor_volume
    d_tumor_volume__dt = self.alpha * tumor_volume * (1 - tumor_volume / kappa) - (self.beta * chemotherapy_drug_concentration *
            tumor_volume) / (self.theta + chemotherapy_drug_concentration) - self.gamma * radiotherapy_dosage * tumor_volume /
            (1 + torch.exp(-self.rho * (radiotherapy_dosage - 1))) - self.eta * chemotherapy_drug_concentration *
            radiotherapy_dosage * tumor_volume / resistance

    # Non-linear decay model for chemotherapy drug concentration
    d_chemotherapy_drug_concentration__dt = chemotherapy_dosage - delta * torch.pow(chemotherapy_drug_concentration, 1.5)

    # Black box residual component
    residuals = self.residual_mlp(torch.stack((tumor_volume, chemotherapy_drug_concentration, chemotherapy_dosage,
            radiotherapy_dosage), dim=1))

    # Combine white box model with residuals
    d_tumor_volume__dt += residuals[:, 0]
    d_chemotherapy_drug_concentration__dt += residuals[:, 1]

    return (d_tumor_volume__dt, d_chemotherapy_drug_concentration__dt)
```
optimized_parameters = {'alpha': 0.016903197392821312, 'beta': 0.5582640171051025, 'gamma': 0.08495774865150452, 'kappa_base':
    1029.802490234375, 'kappa_mod': -2.4545891284942627, 'delta_base': 0.12832798063755035, 'delta_mod': -3.1435782148037106e-05,
    'eta': 0.006932476069778204, 'theta': 13.699580192565918, 'rho': 0.24815633893013, 'zeta': 0.13727830350399017}
###

## H.5    HDTwinGen Ablation No Memory

**Ablation Study**. We also ablate HDTwin by removing its memory, only keeping the last hybrid model it generated. We observe decreased performance as shown in Table 4.

Table 4: HDTwinGen Ablation

| Method | Lung Cancer (with Chemo. & Radio.) $\mathcal{T}_{MSE} \downarrow$ |
|---|---|
| HDTwinGen | 0.0889±0.0453 |
| HDTwinGen-no-memory | 17.6±215 |

## H.6    Evaluating Different LLMs

We performed a complete re-run of our main experiments under the same settings, now using a different LLM within our HDTwinGen framework, GPT-3.5. These new results are included in Table 5. HDTwinGen can operate fully with a less capable LLM model. However, the generated and discovered models' performance correlates to the underlying LLM model's performance as expected.

Table 5: **Ablation of using different LLMs.** Test MSE ($\mathcal{T}_{MSE}$) averaged over ten random seeds. HDTwinGen is capable of using other LLM models, however, the best performance results are provided with better-performing LLMs (e.g. GPT-4). The results are presented with $\pm$ indicating 95% confidence intervals.

| Method | Lung Cancer $\mathcal{T}_{MSE} \downarrow$ | Lung Cancer (with Chemo.) $\mathcal{T}_{MSE} \downarrow$ | Lung Cancer (with Chemo. & Radio.) $\mathcal{T}_{MSE} \downarrow$ | Hare-Lynx $\mathcal{T}_{MSE} \downarrow$ | Plankton Microcosm $\mathcal{T}_{MSE} \downarrow$ | COVID-19 $\mathcal{T}_{MSE} \downarrow$ |
|---|---|---|---|---|---|---|
| HDTwinGen (GPT-3.5) | $2.89e + 03 \pm 6.17e + 03$ | $151 \pm 57.8$ | $46.4 \pm 52.1$ | $298 \pm 19.2$ | $0.0007 \pm 0.000233$ | $56 \pm 28.2$ |
| **HDTwinGen (GPT-4)** | **4.41 ± 8.07** | **0.0889 ± 0.0453** | **0.131 ± 0.198** | **291 ± 30.3** | **2.51e-06 ± 2.2e-06** | **1.72 ± 2.28** |

We also explored the effect of changing the LLMs underlying temperature hyperparameter, here using GPT-4, varying the temperature from 0, 0.7 to 2.0, where we used 0.7 throughout all our experiments (Appendix E). As outlined in Table 6, we observe that HDTwinGen is still able to operate with a different underlying temperature of the LLM.

## H.7    Prompt Ablations with Varying Amounts of Prior Information

We conducted a complete re-run of our main experiments, ablating the prompt, which provides the prior information in the form of a textual prior to HDTwinGen. We provide the ablation results in Table 7. Specifically the prompt as outlined in Appendix E is structured to include separately a

Table 6: **Ablation of varying the LLMs temperature.** HDTwinGen can still operate with different LLM temperatures. Reporting the test prediction MSE ($\mathcal{T}_{MSE}$) averaged over ten random seeds.

| Method | Lung Cancer $\mathcal{T}_{MSE}\downarrow$ | Lung Cancer (with Chemo.) $\mathcal{T}_{MSE}\downarrow$ | Lung Cancer (with Chemo. & Radio.) $\mathcal{T}_{MSE}\downarrow$ | Hare-Lynx $\mathcal{T}_{MSE}\downarrow$ | Plankton Microcosm $\mathcal{T}_{MSE}\downarrow$ | COVID-19 $\mathcal{T}_{MSE}\downarrow$ |
|---|---|---|---|---|---|---|
| HDTwinGen (Temp 0.0) | 8.84±17.8 | 0.353±0.936 | 0.414±1.12 | 258±32.9 | 4.06e-05±0.000102 | 0.0461±2.37 |
| HDTwinGen (Temp 0.7) | **4.41±8.07** | **0.0889±0.0453** | **0.131±0.198** | **291±30.3** | **2.51e-06±2.2e-06** | **1.72±2.28** |
| HDTwinGen (Temp 2.0) | 1.05±2.27 | 0.548±0.698 | 8.43±26.4 | 249±53 | 2.32e-06±3.26e-06 | 0.0447±3.95 |

{system description}, {skeleton code}, and {useful to know} components. Here, prior information is conveyed through the {system description}, which describes the system to be modeled, its features, and their ranges; minor system prior information is also conveyed through the {skeleton code}, as this includes task-specific feature names as input variables. The other components of the prompt (e.g. {useful to know}), do not include any task-specific information, and are there to provide general instructions to make the framework work, such as generate a pytorch model as code in the response.

We ablate these components of the prompt, first by removing the task-specific prior {system description} labeled **HDTwinGen (Partial Context)**; second, by removing all task-specific priors removing both {system description} and {skeleton code} (where we change the feature names to meaningless names such as x1, x2, etc.) labeled **HDTwinGen (No Context)**; third, by removing only the {useful to know} information that helps the framework, such as instructions to decompose the system, and or combine white-box models with black-box models for the white-box model residuals, labeled **HDTwinGen (No Instructions)**. We observe in the tabulated results (Table 7) that HDTwinGen can still operate without any task-specific prior information, however having textual prior aids in generating better-performing models, and partially removing HDTwinGen operation instructions, makes it generate slightly less good models.

Table 7: **Prompt Ablations with Varying Amounts of Prior Information.** Test MSE $\mathcal{T}_{MSE}$ averaged over ten random seeds. HDTwinGen can still operate without any task-specific prior information, however having textual prior aids in generating better-performing models. The results are averaged over ten random seeds, with ± indicating 95% confidence intervals.

| Method | Lung Cancer $\mathcal{T}_{MSE}\downarrow$ | Lung Cancer (with Chemo.) $\mathcal{T}_{MSE}\downarrow$ | Lung Cancer (with Chemo. & Radio.) $\mathcal{T}_{MSE}\downarrow$ | Hare-Lynx $\mathcal{T}_{MSE}\downarrow$ | Plankton Microcosm $\mathcal{T}_{MSE}\downarrow$ | COVID-19 $\mathcal{T}_{MSE}\downarrow$ |
|---|---|---|---|---|---|---|
| HDTwinGen (Partial Context) | 6.77±6.4 | 0.601±1.83 | 0.061±0.159 | 277±54.3 | 3.9e-06±8.87e-06 | 2.3±4.24 |
| HDTwinGen (No Context) | 30.3±50.3 | 2.57±2.44 | 1.52±1.98 | 297±51.9 | 6.91e-06±4.11e-06 | 5.12e+10±1.31e+11 |
| HDTwinGen (No Instructions) | 2.31±2.78 | 0.0933±0.287 | 0.212±0.0487 | 313±63.1 | 0.0016±0.00407 | 17.3±47.7 |
| **HDTwinGen** | **4.41±8.07** | **0.0889±0.0453** | **0.131±0.198** | **291±30.3** | **2.51e-06±2.2e-06** | **1.72±2.28** |

## H.8  Domain-Specific Baselines

We performed a complete re-run of our main datasets using domain-specific white-box baselines, as determined by a human expert, as shown in Table 8. Specifically, for COVID-19 modeling, we fit a SEIR model [84], a Lotka–Volterra model for predator-prey population dynamics (Hare-Lynx & Plankton Microcosm datasets) [85], and a logistic tumor growth model with chemo. & radio. effects modeling. HDTwinGen still models the system most accurately, achieving the lowest test prediction MSE on the held-out test dataset of state-action trajectories.

**SEIR Model for COVID-19 Modeling**. The SEIR model is a compartmental model used in epidemiology to simulate how a disease spreads through a population. It divides the population into four compartments: susceptible (S), exposed (E), infectious (I), and recovered (R). The transitions between these compartments are governed by the following differential equations:

$$\frac{dS}{dt} = -\beta \frac{SI}{N},$$
$$\frac{dE}{dt} = \beta \frac{SI}{N} - \sigma E,$$
$$\frac{dI}{dt} = \sigma E - \gamma I,$$
$$\frac{dR}{dt} = \gamma I,$$

Table 8: **Table 5.** (Test MSE $\mathcal{T}_{MSE}$ averaged over ten random seeds)

| Method | Lung Cancer | Lung Cancer (with Chemo.) | Lung Cancer (with Chemo. & Radio.) | Hare-Lynx | Plankton Microcosm | COVID-19 |
|---|---|---|---|---|---|---|
| Domain Specific Baselines Description | Logistic Tumor Growth | Logistic Tumor Growth (with Chemo.) | Logistic Tumor Growth (with Chemo. & Radio.) | Lotka–Volterra | Multi-species Lotka–Volterra | SEIR |
| Domain Specific Baselines | 904±162 | 200±71.1 | 6.39±0.637 | 346±6.7 | 0.0127±0.00203 | 7.88±0.046 |
| **HDTwinGen** | **4.41±8.07** | **0.0889±0.0453** | **0.131±0.198** | **291±30.3** | **2.51e-06±2.2e-06** | **1.72±2.28** |

where $N$ is the total population (assumed constant), $\beta$ is the transmission rate, $\sigma$ is the rate at which exposed individuals become infectious, and $\gamma$ is the recovery rate. These parameters are crucial for capturing the dynamics of the disease spread and are estimated from data or literature.

**Lotka–Volterra Model for Predator-Prey Dynamics**. The Lotka-Volterra model describes the dynamics of biological systems in which two species interact, one as a predator and the other as prey. The model is represented by a set of first-order, non-linear, differential equations:

$$\frac{dx}{dt} = \alpha x - \beta xy,$$
$$\frac{dy}{dt} = \delta xy - \gamma y,$$

where $x$ and $y$ represent the prey and predator populations, respectively. The parameters $\alpha$, $\beta$, $\gamma$, and $\delta$ denote the prey reproduction rate, the predation rate upon the prey, the predator mortality rate, and the rate at which predators increase by consuming prey, respectively.

**Lotka-Volterra Triple Species Model**. The extended Lotka-Volterra model incorporating a third species involves additional interactions that can represent various ecological relationships such as competition, predation, or mutualism. For the sake of illustration, let's consider a system with two predators and one prey. The model is described by the following set of differential equations:

$$\frac{dx}{dt} = x(\alpha - \beta y - \delta z),$$
$$\frac{dy}{dt} = y(-\gamma + \epsilon x),$$
$$\frac{dz}{dt} = z(-\mu + \nu x),$$

where: $x$ represents the prey population. $y$ and $z$ represent the two predator populations. $\alpha$ is the natural growth rate of the prey in the absence of predation. $\beta$ and $\delta$ are the predation rates of the first and second predators on the prey, respectively. $\gamma$ and $\mu$ are the natural death rates of the first and second predators, respectively, in the absence of the prey. $\epsilon$ and $\nu$ are the growth rates of the first and second predators per unit of prey consumed.

**Logistic Tumor Growth Model with Treatment Effects**. The logistic tumor growth model with chemotherapy and radiotherapy effects incorporates the logistic growth model's capacity to simulate the saturation effect observed in tumor growth, alongside treatment effects. The model can be described as:

$$\frac{dN}{dt} = rN\left(1 - \frac{N}{K}\right) - C(N) - R(N),$$

where $N$ is the tumor cell population, $r$ is the intrinsic growth rate of the tumor, and $K$ is the carrying capacity of the environment. $C(N)$ and $R(N)$ represent the effects of chemotherapy and radiotherapy on the tumor cell population, respectively. These treatment functions are often modeled based on dose-response curves and can vary depending on the specific drugs and radiation doses used.

### H.9 Procedurally Generated Synthetic Model Benchmark

We performed a complete re-run of our main baselines on a new entirely procedurally generated synthetic model benchmark. Specifically, by procedurally generating synthetic models, this allows us to test how HDTwinGen performs when the LLM has never seen such a model. To create diverse synthetic models, we modified the structure of underlying cancer with chemo and radio models to incorporate non-biological random modifications, which include the use of trigonometric operators and division operators. In the following, we provide the exact changes made and the structure of the underlying equation.

**Synthetic 1 (inc. $\gamma \sin(\omega t)$).** Here the underlying equation is

$$\frac{dx(t)}{dt} = \left( \rho \log \left( \frac{K}{x(t)} \right) - \beta_c C(t) - (\alpha_r d(t) + \beta_r d(t)^2) + \gamma \sin(\omega t) \right) x(t)$$

**Synthetic 2 (inc. $-\delta I(t)$)**

$$\frac{dx(t)}{dt} = \left( \rho \log \left( \frac{K}{x(t)} \right) - \beta_c C(t) - (\alpha_r d(t) + \beta_r d(t)^2) - \delta I(t) \right) x(t)$$

**Synthetic 3 (inc. $\log(\frac{K}{x(t)+N(t)})$)**

$$\frac{dx(t)}{dt} = \left( \rho \log \left( \frac{K}{x(t) + N(t)} \right) - \beta_c C(t) - (\alpha_r d(t) + \beta_r d(t)^2) \right) x(t)$$

**Synthetic 4 (inc. $\epsilon \cos(\phi t)$)**

$$\frac{dx(t)}{dt} = \left( \rho \log \left( \frac{K}{x(t)} \right) - \beta_c C(t) - (\alpha_r d(t) + \beta_r d(t)^2) + \epsilon \cos(\phi t) \right) x(t)$$

**Synthetic 5 (inc. $\theta C(t)d(t)$)**

$$\frac{dx(t)}{dt} = \left( \rho \log \left( \frac{K}{x(t)} \right) - \beta_c C(t) - (\alpha_r d(t) + \beta_r d(t)^2) - \theta C(t)d(t) \right) x(t)$$

We observe in the tabulated results in Table 9 that HDTwinGen can still generate models that perform well.

Table 9: **Procedurally Generated Synthetic Model Benchmark**. Test MSE $\mathcal{T}_{MSE}$ averaged over ten random seeds. Reporting the test prediction MSE ($\mathcal{T}_{MSE}$) of the produced system models on held-out test datasets across all synthetic datasets. HDTwinGen achieves the lowest test prediction error. The results are averaged over ten random seeds, with $\pm$ indicating 95% confidence intervals.

| Method | Synthetic 1 (inc. $\gamma \sin(\omega t)$) $\mathcal{T}_{MSE} \downarrow$ | Synthetic 2 (inc. $-\delta I(t)$) $\mathcal{T}_{MSE} \downarrow$ | Synthetic 3 (inc. $\log(\frac{K}{x(t)+N(t)})$) $\mathcal{T}_{MSE} \downarrow$ | Synthetic 4 (inc. $\epsilon \cos(\phi t)$) $\mathcal{T}_{MSE} \downarrow$ | Synthetic 5 (inc. $\theta C(t)d(t)$) $\mathcal{T}_{MSE} \downarrow$ |
|---|---|---|---|---|---|
| DyNODE | 65.9±5.82 | 17.6±15.3 | 12.8±5.32 | 63±4.28 | 15.1±8.64 |
| SINDy | 69±1.87 | 16.2±0.972 | 13.7±0.574 | 68.4±1.45 | 14.2±0.598 |
| ZeroShot | 6.05e+03±3.77e+03 | 1.18e+04±2.25e+04 | 6e+03±4.16e+03 | 3.86e+03±3.73e+03 | 4.56e+03±3.84e+03 |
| ZeroOptim | 56.2±1.96 | 14.5±1.17 | 1.82±0.774 | 56.9±1.82 | 3.07±1.11 |
| HDTwinGen | **54.2±2.55** | **0.0707±0.113** | **0.245±0.377** | **54.8±1.98** | **0.0683±0.0464** |

## H.10  Interpretability Scale, Performance of only White-Box Models

To investigate questions, of how well do the white-box models that HDTwinGen generates perform, we explore an ablation of HDTwinGen where we constrain the generated models to be white-box only, i.e., a mathematical equation with no black-box neural network components. We tabulate this in Table 10, and observe that even when HDTwinGen is constrained to only generate white-box models (**HDTwinGen (Only White-Box)**) it still performs acceptably, indicating that the white-box generated models are modeling well the underlying system when fitted to the dataset.

Table 10: **Interpretability Scale, Performance of only White-Box Models**. Reporting the test prediction MSE ($\mathcal{T}_{MSE}$) of the produced system models on held-out test datasets across all benchmark datasets. The results are averaged over ten random seeds, with $\pm$ indicating 95% confidence intervals.

| Method | Lung Cancer $\mathcal{T}_{MSE} \downarrow$ | Lung Cancer (with Chemo.) $\mathcal{T}_{MSE} \downarrow$ | Lung Cancer (with Chemo. & Radio.) $\mathcal{T}_{MSE} \downarrow$ | Hare-Lynx $\mathcal{T}_{MSE} \downarrow$ | Plankton Microcosm $\mathcal{T}_{MSE} \downarrow$ | COVID-19 $\mathcal{T}_{MSE} \downarrow$ |
|---|---|---|---|---|---|---|
| HDTwinGen (Only White-Box) | 59.4±101 | 4.8±11.8 | 2.42±2.02 | 337±25.4 | 0.000111±0.000125 | 5.92±1.17 |
| HDTwinGen (White-Box & Black-box residuals) | **4.41±8.07** | **0.0889±0.0453** | **0.131±0.198** | **291±30.3** | **2.51e-06±2.2e-06** | **1.72±2.28** |

## H.11  HDTwinGen Flexibly Integrates Expert-in-the-loop Feedback

Experts play an active role in model development in two main ways:

1. **Initial prompt.** Experts can describe the system, and specify modeling goals and task-specific requirements through $\mathcal{S}^{context}$ (Appendix E).

2. **Direct model feedback.** In each iteration of HDTwinGen, the expert can provide direct feedback through $H$ to guide model improvement.

To further demonstrate the flexibility of expert involvement in the modeling process, we supply two additional experiments:

1. Expert specifies instructions to produce a fully white-box model through $\mathcal{S}^{context}$, which is provided in Appendix H.10.

2. Expert provides specific feedback on model improvement during the development process through $H$. We performed this experiment by stopping HDTwinGen during its iterations, providing human expert targeted feedback, and then observing if the subsequent model generated was able to incorporate the feedback by making targeted changes to the underlying model. We confirm that this is the case and provide insight that the underlying LLM is able to interpret the HDTwin code model and selectively change parts. We provide a figure to illustrate this result, as seen in Figure 7.

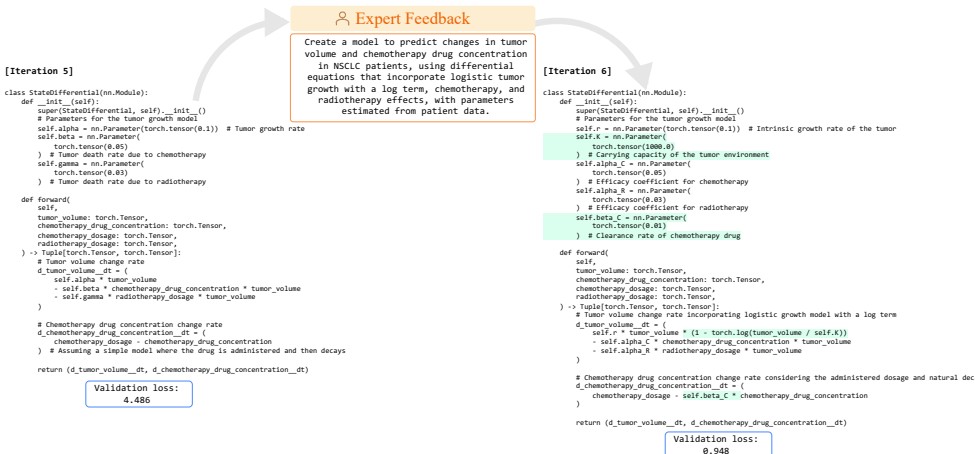

Figure 7: HDTwinGen can flexibly integrate expert-in-the-loop feedback, if it is provided.

## H.12 HDTwinGen Accelerates Model Development and Enhances Performance

We seek to determine the runtimes of HDTwinGen model development compared to human-clock time from a human experiment (Mechanical Turk) experiment where participants are asked to refine models and how performant these models are compared to Bayesian optimization (BO) for a fixed model budget. To assess this we performed the additional experiments of:

1. **Human-driven model development:** Hiring two experienced software engineers (Mechanical Turks) to develop and refine models, providing them with exactly the same prompts that HDTwinGen uses, using a human instead of the LLM in the model development loop.

2. **AutoML:** Hyperparameter tuning (HPT) using for DyNODE and SINDy using BO (HPT search space detailed in Table 11).

We performed these two experiments on the Lung Cancer (with Chemo. & Radio.) dataset, and the results are tabulated in Table 12. Here we present, the time to generate an individual model (which includes generating the model and training time); the test MSE $\mathcal{T}_{MSE}$ after one hour of run-time, and test MSE separately for a budget of only 15 model evaluations.

Analyzing the results provides the following insights:

1. **HDTwinGen Makes Model Development Notably Faster Compared to Human-Clock Time:** HDTwinGen takes an average of 45.56 minutes to complete an experiment using 15

Table 11: **HPT details.** BO HPT search space for Figure 8.

| Method | Hyperparameter Search Space | Best Params |
|---|---|---|
| DyNode learning_rate | [1e-5, 1e-1] | 0.0123 |
| DyNode weight_decay | [1e-5, 1e-1] | 0.00029 |
| DyNode hidden_dim | [32, 1024] | 788 |
| DyNode model_activation | [tanh, silu, ELU] | tanh |
| DyNode model_initialization | [xavier, normal] | normal |
| SINDy polynomial_library_degree | [1, 3] | 2 |
| SINDy polynomial_library_interaction_only | [True, False] | True |
| SINDy threshold | [1e-5, 1e-1] | 0.0194 |
| SINDy alpha | [1e-5, 1e-1] | 0.0015 |

Table 12: **Method performance comparison.** Reporting the time to generate a model (in minutes) and the test MSE ($\mathcal{T}_{MSE}$) after one hour of run-time and with a budget of only 15 model evaluations. Here $^*$ indicates 73.11% of iteration time is consumed by LLM querying.

| Method | Time to generate a model (Minutes) ↓ | Test MSE $\mathcal{T}_{MSE}$ (After one hour of run-time) ↓ | Test MSE $\mathcal{T}_{MSE}$ (Budget of only 15 model evaluations) ↓ |
|---|---|---|---|
| HPO for DyNODE | 0.37 | 1.122 | 2.209 |
| HPO for SINDy | 0.16 | 13.225 | 13.245 |
| Human Experts (Mechanical Turk) | 9.875 | 68.004 | 2.209 |
| HDTwinGen | 3.037$^*$ | **0.072** | **0.072** |

model evaluations/generations, whereas human experts took 148.1 minutes (2 hours and 28 minutes) to generate and iterate 15 models. Therefore, using HDTwinGen is considerably faster than human-clock time, which is also an advantage of HDTwinGen.

2. **HDTwinGen Generates Better Performing Models for a Fixed Model Budget:** HDTwinGen generates better-performing models compared to the human experts and the Bayesian hyperparameter optimization (HPO) of the baselines of DyNode and SINDy, for a fixed budget of 15 model evaluations. We provide a figure showing the best-performing model performance against each generation in a Figure 8. This figure shows that across model evaluations, HDTwinGen still generates models that perform well.

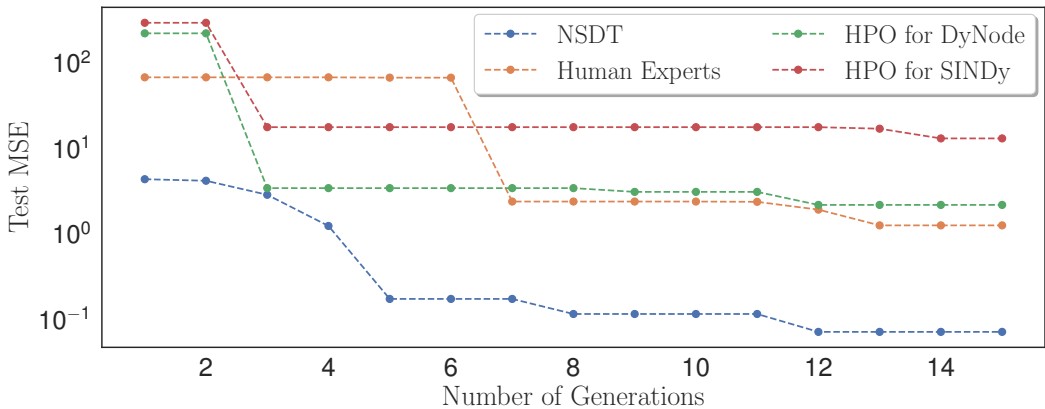

Figure 8: Best-performing model performance against each generation, for setup in Appendix H.12. HDTwinGen Generates Better Performing Models for a Fixed Model Budget.

# I   Hybrid Model Output Examples

These are the final discovered hybrid models generated from our method HDTwin, for each respective environment.

## Cancer (with Chemo & Radio)

```python
class StateDifferential(nn.Module):
    def __init__(self):
        super(StateDifferential, self).__init__()
        # Parameters for the tumor growth model
        self.alpha = nn.Parameter(torch.tensor(0.03))  # Tumor growth rate
        self.beta = nn.Parameter(
            torch.tensor(0.4)
        )  # Tumor death rate due to chemotherapy
        self.gamma = nn.Parameter(
            torch.tensor(0.08)
        )  # Tumor death rate due to radiotherapy
        self.kappa_base = nn.Parameter(
            torch.tensor(1030.0)
        )  # Base carrying capacity of the environment
        self.kappa_mod = nn.Parameter(
            torch.tensor(-2.0)
        )  # Modifier for carrying capacity based on treatment
        self.delta_base = nn.Parameter(
            torch.tensor(0.1)
        )  # Base decay rate of the chemotherapy drug
        self.delta_mod = nn.Parameter(
            torch.tensor(0.0003)
        )  # Modifier for decay rate based on tumor volume
        self.eta = nn.Parameter(
            torch.tensor(0.004)
        )  # Interaction term between chemotherapy and radiotherapy
        self.theta = nn.Parameter(
            torch.tensor(10.0)
        )  # Michaelis-Menten saturation constant for chemotherapy
        self.rho = nn.Parameter(
            torch.tensor(0.3)
        )  # Sigmoid steepness for radiotherapy effect
        self.zeta = nn.Parameter(torch.tensor(0.15))  # Resistance development rate

        # Black box component for capturing residuals
        self.residual_mlp = nn.Sequential(
            nn.Linear(4, 16),
            nn.LeakyReLU(0.1),
            nn.Linear(16, 8),
            nn.LeakyReLU(0.1),
            nn.Linear(8, 2),
        )

    def forward(
        self,
        tumor_volume: torch.Tensor,
        chemotherapy_drug_concentration: torch.Tensor,
        chemotherapy_dosage: torch.Tensor,
        radiotherapy_dosage: torch.Tensor,
    ) -> Tuple[torch.Tensor, torch.Tensor]:
        # Adjusted carrying capacity based on treatment
        kappa = self.kappa_base + self.kappa_mod * (
            chemotherapy_dosage + radiotherapy_dosage
        )

        # Adjusted decay rate based on tumor volume
        delta = self.delta_base + self.delta_mod * tumor_volume

        # Logistic growth model for tumor volume with interaction term and resistance
        resistance = 1 + self.zeta * tumor_volume
        d_tumor_volume__dt = (
            self.alpha * tumor_volume * (1 - tumor_volume / kappa)
            - (self.beta * chemotherapy_drug_concentration * tumor_volume)
            / (self.theta + chemotherapy_drug_concentration)
            - self.gamma
            * radiotherapy_dosage
            * tumor_volume
            / (1 + torch.exp(-self.rho * (radiotherapy_dosage - 1)))
            - self.eta
            * chemotherapy_drug_concentration
            * radiotherapy_dosage
            * tumor_volume
            / resistance
        )

        # Non-linear decay model for chemotherapy drug concentration
        d_chemotherapy_drug_concentration__dt = chemotherapy_dosage - delta * torch.pow(
            chemotherapy_drug_concentration, 1.5
        )

        # Black box residual component
        residuals = self.residual_mlp(
            torch.stack(
                (
                    tumor_volume,
                    chemotherapy_drug_concentration,
                    chemotherapy_dosage,
                    radiotherapy_dosage,
                ),
                dim=1,
            )
        )

        # Combine white box model with residuals
        d_tumor_volume__dt += residuals[:, 0]
        d_chemotherapy_drug_concentration__dt += residuals[:, 1]
```

```python
        return (d_tumor_volume__dt, d_chemotherapy_drug_concentration__dt)
```

## COVID-19

```python
class StateDifferential(nn.Module):
def __init__(self):
    super(StateDifferential, self).__init__()
    # Initialize the parameters for the SEIR model using optimized values
    self.beta = nn.Parameter(torch.tensor(0.2607165277004242))  # Transmission rate
    self.sigma = nn.Parameter(torch.tensor(0.23686641454696655))  # Incubation rate
    self.gamma = nn.Parameter(torch.tensor(0.105068139731884))  # Recovery rate
    self.delta = nn.Parameter(torch.tensor(0.01))  # Death rate

    # Define a neural network for capturing complex patterns
    # Adjust the architecture based on previous iterations
    self.residual_nn = nn.Sequential(
        nn.Linear(4, 12),
        nn.ReLU(),
        nn.Linear(12, 12),
        nn.ReLU(),
        nn.Linear(12, 4)
    )

def forward(self, susceptible: torch.Tensor, exposed: torch.Tensor, infected: torch.Tensor, recovered: torch.Tensor,
    total_population: torch.Tensor) -> Tuple[torch.Tensor, torch.Tensor, torch.Tensor, torch.Tensor]:
    # SEIR model differential equations
    d_susceptible__dt = -self.beta * susceptible * infected
    d_exposed__dt = self.beta * susceptible * infected - self.sigma * exposed
    d_infected__dt = self.sigma * exposed - (self.gamma + self.delta) * infected
    d_recovered__dt = self.gamma * infected - self.delta * infected

    # Calculate residuals using the neural network
    states = torch.stack([susceptible, exposed, infected, recovered], dim=1)
    residuals = self.residual_nn(states)

    # Add residuals to the differential equations
    d_susceptible__dt += residuals[:, 0]
    d_exposed__dt += residuals[:, 1]
    d_infected__dt += residuals[:, 2]
    d_recovered__dt += residuals[:, 3]

    return (d_susceptible__dt, d_exposed__dt, d_infected__dt, d_recovered__dt)
```

## Plankton Microcosm

```python
class StateDifferential(nn.Module):
def __init__(self):
    super(StateDifferential, self).__init__()
    # Define the parameters for the improved model with carrying capacities
    self.alpha = nn.Parameter(torch.tensor(0.022967826575040817))  # Prey growth rate
    self.beta = nn.Parameter(torch.tensor(0.6899635791778564))  # Prey death rate due to predation
    self.gamma = nn.Parameter(torch.tensor(0.15562176704406738))  # Intermediate predator efficiency
    self.delta = nn.Parameter(torch.tensor(0.8135092854499817))  # Top predator efficiency
    # Carrying capacities for each population
    self.K_prey = nn.Parameter(torch.tensor(0.4680666923522949))      # Carrying capacity for the prey population
    self.K_intermediate = nn.Parameter(torch.tensor(0.8180080652236938))  # Carrying capacity for the intermediate predator
            population
    self.K_top = nn.Parameter(torch.tensor(0.4186957776546478))  # Carrying capacity for the top predator population
    # Competition coefficients
    self.sigma = nn.Parameter(torch.tensor(-0.27261480689048767))  # Competition coefficient for prey and intermediate predators
    self.eta = nn.Parameter(torch.tensor(0.06442223489284515))    # Competition coefficient for intermediate predators and top
            predators
    # MLP for residuals with refined architecture
    self.residual_mlp = nn.Sequential(
        nn.Linear(3, 128),
        nn.LeakyReLU(0.01),
        nn.Dropout(0.25),
        nn.Linear(128, 128),
        nn.LeakyReLU(0.01),
        nn.Dropout(0.25),
        nn.Linear(128, 3)
    )

def forward(self, prey_population: torch.Tensor, intermediate_population: torch.Tensor, top_predators_population: torch.Tensor) ->
    Tuple[torch.Tensor, torch.Tensor, torch.Tensor]:
    # Improved differential equations with carrying capacities
    d_prey_population__dt = self.alpha * prey_population * (1 - prey_population / self.K_prey) - self.beta * prey_population *
            intermediate_population
    d_intermediate_population__dt = (self.gamma * prey_population * intermediate_population / (1 + prey_population)) - self.delta *
            intermediate_population * top_predators_population - self.sigma * intermediate_population**2 * (1 -
            intermediate_population / self.K_intermediate)
    d_top_predators_population__dt = -self.gamma * top_predators_population + self.delta * intermediate_population *
            top_predators_population - self.eta * top_predators_population**2 * (1 - top_predators_population / self.K_top)

    # Calculate residuals using MLP
    residuals = self.residual_mlp(torch.stack([prey_population, intermediate_population, top_predators_population], dim=1))

    # Correct the differential equations with the residuals
    d_prey_population__dt += residuals[:, 0]
    d_intermediate_population__dt += residuals[:, 1]
    d_top_predators_population__dt += residuals[:, 2]

    return (d_prey_population__dt, d_intermediate_population__dt, d_top_predators_population__dt)
```

## Hare-Lynx

```python
class StateDifferential(nn.Module):
    def __init__(self):
        super(StateDifferential, self).__init__()
        # Define the parameters for the modified Lotka-Volterra equations
        self.alpha = nn.Parameter(torch.tensor(0.067))  # Prey birth rate
        self.beta = nn.Parameter(torch.tensor(0.005))  # Predation rate
```

```
            self.gamma = nn.Parameter(torch.tensor(0.253))  # Predator death rate
            self.delta = nn.Parameter(torch.tensor(0.0028)) # Predator reproduction rate
            # Additional parameters for carrying capacity and time−dependent effects
            self.kappa = nn.Parameter(torch.tensor(1.0))  # Carrying capacity for hares
            self.omega = nn.Parameter(torch.tensor(2 * torch.pi / 10))  # Frequency of oscillations (10−year period)
            self.phi = nn.Parameter(torch.tensor(0.0))  # Phase shift for oscillations
            # Neural network for capturing residuals
            self.residual_mlp = nn.Sequential(
                nn.Linear(3, 10),
                nn.ReLU(),
                nn.Linear(10, 2)
            )

    def forward(self, hare_population: torch.Tensor, lynx_population: torch.Tensor, time_in_years: torch.
        Tensor) -> Tuple[torch.Tensor, torch.Tensor]:
            # Modified Lotka−Volterra equations with carrying capacity and time−dependent effects
            d_hare_population__dt = self.alpha * hare_population * (1 − hare_population / self.kappa) − self.beta * hare_population *
                lynx_population
            d_lynx_population__dt = −self.gamma * lynx_population + self.delta * hare_population * lynx_population
            # Time−dependent oscillatory component
            time_effect = torch.sin(self.omega * time_in_years + self.phi)
            # Combine white box model with neural network residuals
            residuals = self.residual_mlp(torch.stack((hare_population, lynx_population, time_effect), dim=1))
            d_hare_population__dt += residuals[:, 0]
            d_lynx_population__dt += residuals[:, 1]
            return (d_hare_population__dt, d_lynx_population__dt)
```

# J  HDTwinGen can reason about HDTwins

Worked log output of HDTwinGen of part of a run, running on the Plankton Microcosm dataset. It can reason about structures and parameters.

```
You will get a system description to code a differential equation simulator for.

System Description:```
"Modeling Artificial Tri−Trophic Prey−Predator Oscillations in a Simplified Ecological System

Here you must model the state differential of algae_population, flagellate_population, and rotifer_population; with no input actions
    . This aims to simulate the population dynamics within a simplified tri−trophic ecological system comprising prey (algae),
    intermediate predators (flagellates), and top predators (rotifers). The interactions include direct predation and competition
    for resources, mirroring natural intraguild predation mechanisms.

Description of the variables:
* prey_population: Total count of algae, serving as the primary prey
* intermediate_population: Total count of flagellates, acting as intermediate predators and prey
* top_predators_population: Total count of rotifers, representing top predators

The dataset encapsulates daily population counts across multiple simulated ecosystems over a period of 100 days, allowing for the
    analysis of temporal oscillations and phase lags between species.

Additionally these variables have the ranges of:
* prey_population: [0.095898, 2.469735]
* intermediate_population: [0.008438, 1.500000]
* top_predators_population: [0.030316, 0.739244]

The training dataset consists of 70 time steps, validation and training dataset consists of 15 time steps each.

```

Modelling goals:```
* The parameters of the model will be optimized to an observed training dataset with the given simulator.
* The observed training dataset has very few samples, and the model must be able to generalize to unseen data.
```

Requirement Specification:```
* The code generated should achieve the lowest possible validation loss, of 1e−6 or less.
* The code generated should be interpretable, and fit the dataset as accurately as possible.
```

Skeleton code to fill in:```
class StateDifferential(nn.Module):
    def __init__(self):
        super(StateDifferential, self).__init__()
        # TODO: Fill in the code here

    def forward(self, prey_population: torch.Tensor, intermediate_population: torch.Tensor, top_predators_population: torch.Tensor)
        −> Tuple[torch.Tensor, torch.Tensor, torch.Tensor]:
        # TODO: Fill in the code here
        return (d_prey_population__dt, d_intermediate_population__dt, d_top_predators_population__dt)
```

Useful to know:```
* You are a code evolving machine, and you will be called 20 times to generate code, and improve the code to achieve the lowest
    possible validation loss.
* The model defines the state differential and will be used with an ODE solver to fit the observed training dataset.
* You can use any parameters you want and any black box neural network components (multi−layer perceptrons); however, you have to
    define these.
* It is preferable to decompose the system into differential equations (compartments) if possible.
* You can use any unary functions, for example log, exp, power etc.
* Under no circumstance can you change the skeleton code function definitions, only fill in the code.
* The input tensors are vectors of shape (batch_size).
* Use initially white box models first and then switch to hybrid white and black box models for the residuals, only after no further
    best program iteration improvement with white box models.
* Make sure your code follows the exact code skeleton specification.
* Use PyTorch.
```

Think step−by−step, and then give the complete full working code. You are generating code for iteration 0 out of 20.

MainProcess| 2024−01−29 00:02:01,082,82 multiprocessing INFO [User]
```

You generated the following code completions, which then had their parameters optimized to the training dataset. Please reflect on how you can improve the code to minimize the validation loss to 1e−6 or less. The code examples are delineated by ###.

Here are your previous iterations the best programs generated. Use it to see if you have exhausted white box models, i.e. when a white box model repeats with the same val loss and then only add black box models to the white box models:```
Iteration 0. Best Val Loss: 0.012224067933857441. Model description: white box model
Iteration 1. Best Val Loss: 0.003129902994260192. Model description: white box model with additional parameters for competition and carrying capacity, and improved differential equations to better capture the tri−trophic dynamics.
```

Here are the top code completions so far that you have generated, sorted for the lowest validation loss last:```

Val Loss: 0.0122 (Where the val loss per dimension is prey_population val loss: 0.0316, intermediate_population val loss: 2.13e−05, top_predators_population val loss: 0.00505) Iteration: 0
###
```
import torch
import torch.nn as nn
from typing import Tuple

class StateDifferential(nn.Module):
    def __init__(self):
        super(StateDifferential, self).__init__()
        # Define the parameters for the Lotka−Volterra equations
        self.alpha = nn.Parameter(torch.rand(1))  # Prey growth rate
        self.beta = nn.Parameter(torch.rand(1))  # Prey death rate due to predation
        self.gamma = nn.Parameter(torch.rand(1)) # Predator growth rate due to predation
        self.delta = nn.Parameter(torch.rand(1)) # Predator death rate

    def forward(self, prey_population: torch.Tensor, intermediate_population: torch.Tensor, top_predators_population: torch.Tensor)
            -> Tuple[torch.Tensor, torch.Tensor, torch.Tensor]:
        # Lotka−Volterra equations
        d_prey_population__dt = self.alpha * prey_population − self.beta * prey_population * intermediate_population
        d_intermediate_population__dt = self.gamma * prey_population * intermediate_population − self.delta *
            intermediate_population * top_predators_population
        d_top_predators_population__dt = −self.gamma * top_predators_population + self.delta * intermediate_population *
            top_predators_population

        return (d_prey_population__dt, d_intermediate_population__dt, d_top_predators_population__dt)
```
optimized_parameters = {'alpha': 0.10977201908826828, 'beta': 0.68703693151474, 'gamma': 0.12234053760766983, 'delta':
    0.8221619725227356}
###

Val Loss: 0.00313 (Where the val loss per dimension is prey_population val loss: 0.00776, intermediate_population val loss: 2.31e
    −05, top_predators_population val loss: 0.00161) Iteration: 1
###
```
import torch
import torch.nn as nn
from typing import Tuple

class StateDifferential(nn.Module):
    def __init__(self):
        super(StateDifferential, self).__init__()
        # Define the parameters for the improved model
        self.alpha = nn.Parameter(torch.tensor(0.10977201908826828))  # Prey growth rate
        self.beta = nn.Parameter(torch.tensor(0.68703693151474))  # Prey death rate due to predation
        self.gamma = nn.Parameter(torch.tensor(0.12234053760766983)) # Predator growth rate due to predation
        self.delta = nn.Parameter(torch.tensor(0.8221619725227356)) # Predator death rate
        # Additional parameters for competition and carrying capacity
        self.sigma = nn.Parameter(torch.rand(1)) # Competition coefficient for prey and intermediate predators
        self.eta = nn.Parameter(torch.rand(1))   # Competition coefficient for intermediate predators and top predators
        self.K = nn.Parameter(torch.rand(1))     # Carrying capacity for the prey population

    def forward(self, prey_population: torch.Tensor, intermediate_population: torch.Tensor, top_predators_population: torch.Tensor)
            -> Tuple[torch.Tensor, torch.Tensor, torch.Tensor]:
        # Improved differential equations
        d_prey_population__dt = self.alpha * prey_population * (1 − prey_population / self.K) − self.beta * prey_population *
            intermediate_population
        d_intermediate_population__dt = self.gamma * prey_population * intermediate_population − self.delta *
            intermediate_population * top_predators_population − self.sigma * intermediate_population**2
        d_top_predators_population__dt = −self.gamma * top_predators_population + self.delta * intermediate_population *
            top_predators_population − self.eta * top_predators_population**2

        return (d_prey_population__dt, d_intermediate_population__dt, d_top_predators_population__dt)
```
optimized_parameters = {'alpha': −0.013438135385513306, 'beta': 0.6186316609382629, 'gamma': 0.36335352063179016, 'delta':
    0.6213762760162354, 'sigma': 0.47441932559013367, 'eta': −0.43212178349494934, 'K': 0.3493500351905823}
###

```

Please reflect on how you can improve the code to fit the dataset as accurately as possible, and be interpretable. Think step−by−step. Provide only actionable feedback, that has direct changes to the code. Do not write out the code, only describe how it can be improved. Where applicable use the values of the optimized parameters to reason how the code can be improved to fit the dataset as accurately as possible. This is for generating new code for the next iteration 2 out of 20.

MainProcess| 2024−01−29 00:02:01,082,82 multiprocessing INFO [Assistant] To improve the code for iteration 2, consider the following actionable feedback:

1. **Incorporate Resource Limitation for Intermediate Predators and Top Predators**: The current model includes a carrying capacity term (K) for the prey population but does not account for resource limitation for intermediate predators and top predators. Introduce carrying capacity parameters for these populations to model their growth saturation.

2. **Refine Interaction Terms**: The interaction terms currently use simple multiplicative factors between populations. Consider more complex interactions that may include nonlinear effects or functional responses, such as a Holling Type II functional response, to better capture the dynamics of predation.

3. **Adjust Negative Growth Rate**: The optimized parameter 'alpha' for the prey growth rate has a negative value, which is biologically unrealistic as it would imply the prey population decreases in the absence of predators. Ensure that the growth rate parameters are constrained to be positive during optimization.

4. **Review Competition Coefficients**: The optimized parameters 'sigma' and 'eta' for competition are positive and negative, respectively. While 'sigma' being positive is consistent with competition, 'eta' being negative is not typical for a competition coefficient. Re-evaluate the competition terms in the equations to ensure they are biologically meaningful and constrain the coefficients to be non-negative during optimization.

5. **Introduce Density-Dependent Mortality for Top Predators**: The model currently does not include a term for density-dependent mortality of top predators. Add a term to the differential equation for top predators that accounts for intraspecific competition or other density-dependent mortality factors.

6. **Parameter Initialization Based on Optimized Values**: Use the optimized parameter values from iteration 1 as initial values for the parameters in iteration 2. This can provide a better starting point for the optimization process.

7. **Consider Time Delays**: In real ecosystems, there can be time delays between predation and changes in population sizes due to factors like gestation or maturation periods. If the validation loss does not improve significantly with the above changes, consider adding time delay terms to the model.

8. **Hybrid White and Black Box Model**: If the validation loss does not improve below the threshold of 1e−6 with the above white box model improvements, consider introducing a black box component, such as a multi-layer perceptron, to capture the residuals of the white box model. This hybrid approach can potentially model complex interactions that are not captured by the white box differential equations.

By implementing these changes, the model should become more biologically realistic and may fit the dataset more accurately, potentially achieving a lower validation loss.

MainProcess| 2024−01−29 00:02:01,082,82 multiprocessing INFO [User]
Please now regenerate the code function, with the aim to improve the code to achieve a lower validation error. Use the feedback where applicable. You are generating code for iteration 2 out of 20 total iterations. When generating code if you are unsure about something, take your best guess. You have to generate code, and cannot give an empty string answer.

Please always only fill in the following code skeleton:```
class StateDifferential(nn.Module):
    def __init__(self):
        super(StateDifferential, self).__init__()
        # TODO: Fill in the code here

    def forward(self, prey_population: torch.Tensor, intermediate_population: torch.Tensor, top_predators_population: torch.Tensor)
        -> Tuple[torch.Tensor, torch.Tensor, torch.Tensor]:
        # TODO: Fill in the code here
        return (d_prey_population__dt, d_intermediate_population__dt, d_top_predators_population__dt)
```

You cannot change the code skeleton, or input variables.

