# OpenReview forum: "Automatically Learning Hybrid Digital Twins of Dynamical Systems"
_NeurIPS.cc/2024/Conference — NeurIPS 2024 spotlight_

### Official Review · Reviewer_jD19 · 2024-07-10

**Soundness:** 3
**Presentation:** 4
**Contribution:** 3
**Rating:** 8
**Confidence:** 4

**Summary:**

The paper presents a neurosymbolic approach to model dynamical systems based on the usage of LLMs and gradient-based optimization. The model perform competitively against the reported baselines.

Firstly, the human modeller is required to define the problem, priors and target metrics in text. Then the first LLM generates python models that are optimized with SGD, subsequently a second LLM distill the results and pass it onto the first LLM which generates a new set of candidate models to be trained again.

Performance is reported on synthetic data from different models. Proposed model outperform the reported baselines.

**Strengths:**

- The model allows to integrate domain knowledge by using LLM

- The proposed model outperform reported baselines.

- Paper is very well written.

- Annex incorporate comprehensive information about the experiments.

**Weaknesses:**

- It is unclear whether the computational budget of the baselines (in terms of $\mathcal{L}(f_{\theta, \omega}(o), D_{\text{train}})$ evaluations) is similar to that of the proposed method.


- No code is provided.

- The work relies on a closed-sourced model (GPT-4) whose performance also depends on the specific checkpoints used.

**Questions:**

- Line 55-56: since the work is only shown in the additive case, I suggest you mention this case here also as only showing compositional case results a bit confusing.

- Line 137-139: I find it confusing stating that the outer loss measures generalization. As far as I understand, the outer loss is responsible for model specification, whether this values captures generalization will the depend on whether it's evaluated on the validation dataset, not due to the fact that it is the outer loss.

- I'm unsure whether all the baselines are synthetic data or only some of them. Could you please clarify this?

- In terms of parameter size, how big are the found models $f_{\theta, w(\theta)}$ and how big are the baselines models? Both for the algebraic and neural part.

- Line 188: doesn't $P^{(g)}$ contain $w(\theta)$? If not, could you elaborate on why?

- Why are two LLMs needed? Couldn't a single one do both steps? Have you tried this?

- Suggestion: It'd be interesting to see how the algebraic expression evolve over time during training.

**Limitations:**

- The model adds a substantial amount of complexity with respect to alternative modelling approaches like the reported baseline transformers. It's unclear that the performance improvements makes it up for the added complexity of the method.

- Approach is evaluated against simulated data. If one of the benefit of the approach is that it can handle small datasets, and generalize well, why not using a real dataset rather than synthetic data? This way benchmarks against SoTA on those benchmark would be standardized and therefor more meaningful that self-reported baselines. E.g. reported results for SINDy are using polynomials of order 2, this choice is not justified in the paper. Or, transformers perform similarly to paper's approach (within standard error), and it is not possible to know whether the choice of training hyper-parameters.

- Like other symbolic approaches, the paper's approach still requires a human to provide modelling priors to the system, hence the claim of line 79 "a new approach to automatically learn digital twins" is a bit weaken.

- Besides, Table 10, the explainability potential of the model remains under-investigated.

---

> ### Author Rebuttal · Authors · 2024-08-07
>
> *We appreciate the reviewer’s thorough evaluation and positive feedback.*
>
> ---
> ## [P1] Clarifying computational budget
> Thank you for raising this question. To clarify:
> 1. All neural baselines and each HDTwinGen model use identical training settings (2000 epochs, 1000 batch size), consuming the same number of function evaluations per model.
> 2. HDTwinGen evolves one new model every generation for 20 generations, using a higher total computational budget.
> 3. Exact evaluation counts are provided in the **Questions** response below.
>
> **Empirical comparisons with comparable budgets:** We have included additional results comparing performance against DyNODE and SINDY given equivalent budgets. This additional budget is used towards HPT, where each baseline underwent 15 iterations of HPT, matching HDTwinGen's 15 evolutionary iterations. Results provided in **[A3]** demonstrate HDTwinGen's superior performance, highlighting the performance benefits of evolving model specification beyond HPT.
>
> **Conceptual differences:** HDTwinGen automates specification and parameter design of hybrid digital twins, unlike neural baselines that only optimize expert-specified models' parameters. This automation accelerates and scales model development, improving cost- and time- efficiency. The increased computational complexity trades off with higher automation levels and the potential for better-performing models.
>
> **Actions taken:** We now include the above in App H.
>
> ---
>
> ## [P2] Code release
> Upon acceptance of the paper, we will release the code, accompanied by extensive reproducibility instructions in App E, to ensure reproducibility.
>
> ---
>
> ## [P3] Understanding reliance on underlying LLM
>
> We appreciate this comment and agree that the performance of HDTwinGen depends on the capabilities of the underlying model. To understand how the algorithm scales with different models, we reported the performance of HDTwinGen using LLMs of varying capabilities (App H6). We observed that performance correlated with the capabilities of the underlying LLMs, supporting the hypothesis that HDTwinGen's performance will improve as the underlying models advance.
>
> **Actions taken:** We now include this in S8 of the paper.
>
> ---
>
> ## Questions
>
> * **Composition operator:** We have revised L55-56 to mention that we focus on additive composition.
> * **Bilevel objective:** Your understanding is correct. The upper-level objective concerns model specification (evaluated on the validation set), while the lower-level objective concerns parameterization performance (evaluated on the training set). This has been clarified in L137-139.
> * **Additional details on benchmark:** The three variants of the Lung Cancer dataset are synthetically generated from PKPD models, while the COVID-19 dataset is generated using an agent-based COVASIM simulator. The Plankton Microcosm (three species) and Hare-Lynx (two species) datasets are real-world ecological datasets. The Plankton dataset is measured in laboratory replication experiments, whereas the Hare-Lynx dataset is measured outdoors.
> * **Parameter count:** On Lung Cancer:
>
> |Baseline|Parameter Count|Seconds per Epoch|# Function (Epoch) Evals|
> |---|---|---|---|
> |DyNODE|33,922|0.02|2,000|
> |SINDy|13|0.01|2,000|
> |RNN|569,002|0.41|2,000|
> |Transformer|2,558,348|0.36|2,000|
> |HDTwinGen|245|0.01|40,000|
>
> * **$\omega(\theta)$ in $\mathcal{P}^{(g)}$:** Your understanding is correct, $\mathcal{P}^{(g)}$ is the set of optimized models and includes the optimized parameters $\omega(\theta)^*$. This has been revised in L188-L190.
> * **Clarification on modeling workflow:** In our framework, the modeling process is divided into two subtasks: model generation (the modeling agent) and model evaluation (the evaluation agent). Each subtask has distinct instructions and performs different roles. Although there are two steps, both are implemented using a single LLM with different prompts and memory for each task. To clarify this, we will describe our system as a 'multi-step' agent rather than a multi-agent workflow. The main distinction is that multi-agent frameworks (e.g., AutoGPT, MetaGPT) involve dynamic interactions and task allocation, whereas our workflow is sequential and fixed. Thank you for this suggestion, which has improved the clarity and presentation of our work.
> * **Evolution of mechanistic component:** We direct you to App H4, where we included the models returned by HDTwinGen in each generation and annotated it with some interpretations of the evolution (including mechanistic and neural components).
>
> ---
>
> ## Limitations
>
> * **Complexity and performance tradeoff:** We addressed this concern in detail in **[P1]**. As a novel automated hybrid design strategy, HDTwinGen's computational complexity is indeed higher than manual model building, similar to other automated approaches (eg AutoML, NAS). However, this complexity is justified by: improved model performance, and enhanced scalability in model development, potentially reducing overall cost and time in modeling. This is further supported by our new results (**[A3]**), highlighting that given comparable budgets, HDTwinGen discovers better-performing models.
> * **Standardized comparison:** Please see our response above clarifying synthetic vs real-world datasets. Regarding your comment about the performance benefits of HDTwinGen vs HPT of neural baselines, we addressed this with additional empirical results with comparable computational budgets, which we discussed in **[P1]**.
> * **Sharpening claims:** We have refined our claim in L79 to "a new approach to automatically design digital twins given human modeling priors."
> * **Explainability:** While we have provided some preliminary interpretations of discovered models in App H4 and App H10, we agree that this is an important future direction. We have highlighted this by revising Section 8.
>
> ---
> *We hope the reviewer’s concerns are addressed and they will consider updating their score. We welcome further discussion.*

---

> > ### Comment · Reviewer_jD19 · 2024-08-09
> >
> > Thank you for the clarifications and the work put on getting the new baselines metrics. I think your work makes for a nice contribution, I will update the rate accordingly.

---

> > > ### Author Response · Authors · 2024-08-11
> > > **Thank you**
> > >
> > > We appreciate your insightful feedback, which has been crucial in enhancing the quality of our submission. We are delighted to have successfully addressed your concerns and grateful for your constructive input throughout this process.

---

### Official Review · Reviewer_8ECc · 2024-07-12

**Soundness:** 3
**Presentation:** 3
**Contribution:** 3
**Rating:** 6
**Confidence:** 4

**Summary:**

This paper presents a LLM-powered evolutionary multi-agent algorithm to automate the composition of hybrid digital twins of dynamical systems. Experiments were conducted on several datasets including variants of PKPD model synthesized data, simulated covid-19 data, simulated plankton miscrocosm data, and a real-data of hare and lynx populations. Results versus existing neural ODE, SINDy, RNN, and Transformer demonstrated favorable performance especially in terms of generalizability, sample efficiency, and evolvability of the hybrid model.

**Strengths:**

Hybrid models are gaining increasing importance for combining white- and black-box modeling. LLM-driven systems for automating hybrid model design, composition, and optimization are critical for enabling the practical adoption of these complex models. The proposed work is thus highly novel and of high potential impact.

The experimental evaluation considered several interesting datasets and targeted on several important desiderate of hybrid models.

**Weaknesses:**

While focused on hybrid models, the writing of the paper is missing important related works in recent hybrid models, such as physics-integrated VAE and APHYNITY, in both discussion of related works, and experimentation.

In experiments, the baselines of the works are primarily black-box models (NODE, RNN, Transformers). As the focus of the paper is the automated optimization of hybrid models, it’d be important to 1) include existing hybrid-models as baselines, and 2) demonstrate the benefits of automated design of these hybrid models (vs. the current approach for optimizing these hybrid models where both the white- and black-box components are predefined and only their parameters optimized).

The writing of the paper overall is high-level lacking necessary details for properly understanding and assessing the paper, potentially due to the complexity of the method. For instance, methodologically, it was not clear what is the search space of the hybrid model, for either the white-box or the black-box components; experimentally, it was not clear what the hybrid model is trained to do, what is the rough design of the hybrid components, and the MSE metric is being evaluated on what.

The performance improvements reported (e.g., Table 1) overall seems to be associated with very large standard deviation (in most cases are larger than the mean). Its gain over transformer based approaches is thus not clear given this large fluctuation and the increased complexity of the method.

Most of the experimental data seem to be low in dimension (at each time frame). Please clarify.

**Questions:**

It’d be helpful for the authors to clarify the relation of the presented work with existing hybrid models, and how it can benefit with the optimization of these hybrid models.

It’d be helpful if the authors could add details about the search space of the hybrid models, both in the general methodological settings and in each experimental dataset.

For each experimental dataset considered, it’d be helpful for the authors to add details about the design of the hybrid model (what white box components, what black box components, what search space, to generate what output for what task).


Clarification on the “spatial” dimension of the experimental datasets would be appreciated.

**Limitations:**

The authors provided adequate discussion about the limitation and future work for the presented work.

---

> ### Author Rebuttal · Authors · 2024-08-07
>
> *We thank the reviewer for their thoughtful and helpful review. We’re glad that the reviewer finds our approach to be both highly novel and of high potential impact, though we agree that the addition of existing hybrid models improves the paper significantly.*
>
> ---
> ## [P1] Extending the literature review
> We appreciate the references to related works, which consider hybrid models combining mechanistic equations with neural components. **[R1]** integrates prior ODE/PDE knowledge into a hybrid model, using specialized regularization to penalize the neural component's information content. **[R2]** investigates hybridization in the latent space, employing semantically grounded expert latent variables and neural latent variables that are regularized to reduce divergence from the expert component.
>
> Our work differs significantly in both motivation and methodology:
> * **Motivational difference:** Our method introduces an automated framework to jointly optimize hybrid model specification (evolving both mechanistic components and neural architecture) *and* its parameters. This contrasts with related works, where experts specify the hybrid model design and optimization is limited to parameters.
> * **Methodological novelty:** Our approach uniquely integrates LLMs within an evolutionary framework for automated design, guided by expert-provided task context and data-driven feedback. This leverages LLMs' capabilities in symbolic manipulation, contextual understanding, and learning, enabling the exploration of a vast combinatorial space of hybrid models previously infeasible with standard techniques.
>
> **Actions taken:** *(1)* Extended discussions of related works **[R1,R2]** in L269-280. *(2)* Introduced APHYNITY, an existing hybrid model, as an additional baseline in the global response **[A1]**, demonstrating HDTwinGen's superior performance.
>
> ---
>
> ## [P2] Demonstrating the benefits of automated design of existing hybrid models
> We agree and have added an experiment demonstrating HDTwinGen's ability to further optimize a human-specified APHYNITY model. Specifically, we seed the process with an expert-designed hybrid model (combining logistic tumor growth model **[R7]** with three-layer MLP). The evolution process is visualized in **[A2]** (in the PDF), showing HDTwinGen further evolving the model specification, resulting in improved performance over the initial model specification.
>
> Our automated approach offers substantial benefits: it enables model development at scale, significantly reducing the time and cost compared to human-driven development, involving humans only at key moments (e.g., providing initial context or iterative feedback if desired). Importantly, our approach has the potential to uncover novel and effective hybrid model designs that might elude human designers.
>
> **Actions taken:** *(1)* Added additional results demonstrating HDTwinGen's ability to further evolve an expert-specified APHYNITY model in **[A2]**, *(2)* Included discussion on benefits of automated design in S8.
>
> ---
>
> ## [P3] Clarifications on the algorithm
> Thank you. We highlight that low-level details are already contained within App E,F,G. Here we clarify each aspect:
>
> * **Search spaces:** We do not impose predefined restrictions on mechanistic models (e.g., primitive/terminal sets in symbolic regression). We also do not specify any architecture primitives (e.g., in neural architecture search) for neural components. The search space is implicitly constrained by the symbolic language, as the evolved model must be valid/executable PyTorch python code.
> * **Hybrid model:** Our hybrid model considers an additive composition of mechanistic and neural components of the form $dx(t)/dt = f(x(t))$, and $f = f_{neural} + f_{mechanistic}$. Evolution optimizes the specification of both mechanistic and neural components, and corresponding parameters.
> * **Training/evaluation and metrics:** The hybrid model is trained jointly to minimize the training MSE loss (Eq 5, App F) using the Adam optimizer. Model evaluation is based on the val MSE and val MSE per component (corresponding to val loss per state dimension, Eq 6).
>
> **Actions taken:** We have highlighted App E more prominently in the main paper.
>
> ---
>
> ## [P4] Understanding the variability of results
> **Variability.** Thank you for your observation regarding performance variability in Table 1. As HDTwinGen optimizes both model specification and parameters, this is equivalent to searching in a much larger hypothesis space compared to baselines (that only search in the parameter space). The variability stemming from model specifications is also evident in the higher std dev of ZeroOptim results (optimized model based on zero-shot LLM generated specifications). Additionally, despite higher variability, HDTwinGen-discovered models exhibit several key strengths: superior OOD performance (Table 2), sample efficiency (Fig 2) and easier evolvability to changing dynamics (Fig 4).
>
> **Actions taken:** Updated manuscript to discuss variability and the trade-offs of automation explicitly in S8.
>
> **Empirical comparisons with comparable budgets:** We have included additional results comparing performance against DyNODE and SINDY given equivalent budgets. This additional budget is used towards HPT, where each baseline underwent 15 iterations of HPT, matching HDTwinGen's 15 evolutionary iterations. Results provided in **[A3]** (in additional PDF) demonstrate HDTwinGen's superior performance, highlighting the performance benefits of evolving model specification beyond HPT.
>
> **Actions taken:** Included empirical comparisons given comparable computational budget in App H.
>
> ---
>
> ## [P5] Clarifying spatial dimensions
> The spatial dimensions of considered benchmarks are Cancer PKPD (4), COVID-19 (4), Plankton-Microcosm (3), Hare-Lynx (2).
>
> ---
> *We hope that most of the reviewer’s concerns have been addressed and, if so, they would consider updating their score. We’d be happy to engage in further discussions.*

---

> > ### Comment · Reviewer_8ECc · 2024-08-08
> >
> > Thanks for a thorough rebuttal which has clarified my main concerns -- the expansion of literature review and the addition of comparison to existing hybrid models is highly appreciated. I will raise my rating.

---

> > > ### Author Response · Authors · 2024-08-09
> > > **Thank you**
> > >
> > > Thank you for your valuable feedback. We are glad to have addressed your concerns and appreciate your insights, which have significantly enhanced the quality of our work.

---

### Official Review · Reviewer_YMit · 2024-07-12

**Soundness:** 2
**Presentation:** 3
**Contribution:** 2
**Rating:** 7
**Confidence:** 3

**Summary:**

This paper describes a method to improve digital twins (DTs) with an evolutionary and human driven dynamics, aided by an LLM. The process is meant to optimize hybrid models with both human-driven and computer search-based optimization.The empirical study uses six datasets from the medicine domain. Suitable baselines and ablation tests are also shown. The authors show that a hybrid approach could lead to superior performance with respect to traditional approaches.

**Strengths:**

I have identified the following strengths:
- The paper addresses an important problem with DT: the need for continuous optimization in front of unseen condition to main the fidelity and effectiveness of DT
- The method suggests that significant autonomy in the optimization could be achieved by means of an evolutionary process, while still retaining the valuable contribution of human domain knowledge for the setting of the initial parameters and context.
- The approach is general and could be applied to a variety of problem domains.
- The approach combines LLMs with optimization techniques such as evolutionary search, producing an advanced algorithm with high potential in the successful use and deployment of DT.
- The paper provide significant details for reproducibility in the Appendix.

**Weaknesses:**

1. I don't think that having two agents: a modelling and an evaluation agent makes the system a "multi-agent" system. I find it confusing. While this is not technically a weakness, I feel that this system should not be considered a multi-agent system.
2. As the authors also state evolutionary computation could be computationally expensive and hide several levels of complexity, e.g., how thorough should the fitness evaluation be, how many individuals in a population and how many generations, what kind of mutation is applied and what intensity. I feel that all such aspects are not well described in the main paper despite the evolutionary part of the algorithm is a fundamental part of the algorithm.
3. Following-up from the previous point, more discussion on the computational complexity of the algorithm in the main paper could be beneficial.
4. The paper could improve the clarity in relation to which domains, or rather, which domain characteristics are more suitable to the proposed algorithm. It is likely that the algorithm might not perform equally well in different domains
5. Despite a fair review of existing related methods in section 5, the paper does not state too clearly what are the core novelty aspects that are being introduced with respect to the state of the art: I can infer part of them by carefully examining the cited papers, but it would be better if the authors could highlight the main novelty in a more technical way with specific references to the most similar papers from the literature.

**Questions:**

In section 4, the paragraph Evolutionary Optimization overview: is it possible to provide more details in relation to the computational time and various aspects as per my point 2 above (weakness) in relation to the evolutionary search?

In the limitations below, I have two questions that can be answered.

**Limitations:**

The limitations section highlights relying on human input, the knowledge of the LLM,  and the limitation to continuous time systems. However, I feel that the following other limitations should be considered:
- how can we guarantee that the LLMs, regardless of its knowledge, can provide correct information? Is there a way to test or assess the performance of the LLM?
- The variability and stochasticity of the evolutionary search could imply that the algorithm might not be consistent across multiple runs. Can the authors speculate on how to measure variability and uncertainty introduced by the evolutionary search?

Both points and questions above are particular relevant in critical domains and scenarios such as the medical scenarios used for demonstration in the paper.

I think a discussion on the implications and ethical aspects of using LLMs, human domain knowledge and evolutionary search all in combination for a wide range of domain, e.g. medical domain, is necessary.

---

> ### Author Rebuttal · Authors · 2024-08-07
>
> *We thank the reviewer for their constructive feedback. We’re glad the reviewer finds the approach general and of high potential.*
>
> ---
> ## [P1] Clarifying "Multi-agent" terminology
> We appreciate your comment on our system's classification as "multi-agent". Upon reflection, we agree that our framework doesn't fully embody all characteristics typically associated with LLM multi-agent systems. While our approach incorporates two distinct components (modeling and evaluation) with specialized roles, they are applied sequentially, lacking dynamic, real-time inter-agent communication and autonomous behavior adjustment in multi-agent systems **[R8, R9]**.
>
> We reclassified our approach as a "multi-step" agent, which more accurately reflects its nature:
> * A modeling step that generates and optimizes new model specifications.
> * An evaluation step that assesses generated models reflects on requirements, and provides targeted improvement feedback.
>
> **Actions taken:** We have now updated this in the paper. Thank you for helping us refine terminology and more precisely position our work.
>
> ---
> ## [P2] Analyzing computational complexity
> Thank you for raising this question. Allow us to clarify the evolutionary process and analyze the computational complexity.
>
> **Evolutionary process:**
> * Each iteration proposes and optimizes one new model specification, conditioned on the top-K previous models
> * The population size N=g in the $g^{th}$ iteration
> * Only the new model's fitness is evaluated on the validation set
> * The selection step retains the top-K models
> * Process repeats for G iterations (G=20, K=16 in our experiments)
>
> **Computational complexity.** We introduce lower-case notation to indicate constant time parameters:
> * Evolution/model generation (LLM): $\mathcal{O}(c)$ (LLM inference)
> * Parameter optimization: $\mathcal{O}(d)$ (training on $\mathcal{D}_{train}$, in practice, depends on model/dataset complexity)
> * Fitness evaluation: $\mathcal{O}(e)$ (on $\mathcal{D}_{val}$)
> * Model evaluation (LLM): $\mathcal{O}(f)$ (LLM inference for improvement suggestions)
> * Selection: $\mathcal{O}(NlogN)$ (top-K)
>
> The complexity per step is $\mathcal{O}(c+d+e+f+NlogN)$, with total complexity for G generations: $\mathcal{O}(G*(c+d+e+f+NlogN))$. We note that constant complexities vary with different datasets and models. For practical insight, we report average wall-clock times on the Lung Cancer dataset: $c=20s$, $d=12s$, $e=0.004s$, $f=20s$.
>
> **Actions taken:** Added complexity analysis and wall-clock times in App H.
>
> ---
> ## [P3] Discussing domains suitable for the algorithm
> We appreciate this suggestion and clarify the domain characteristics our algorithm is designed for:
> * Continuous-time dynamical systems with a) Markov property and b) continuous time evolution.
> * Scenarios where hybrid models are likely to outperform purely mechanistic or neural models: a) partial knowledge of underlying dynamics exists, with some non-negligible unclear aspects, b) limited or unevenly distributed empirical data across the state space.
> * Complex model design spaces with multiple hypotheses: where automated exploration can help experts scale model development, review and discover a wider range of potential designs, improving cost- and time-efficiency of model development.
>
> Our experiments focused on systems with partially understood mechanisms and limited observational data. We observed that our algorithm discovered performant hybrid models, leading to more accurate model dynamics, OOD generalization, and improved evolvability with changing conditions.
>
> **Actions taken:** Included a version of this discussion in S8 and in App A.
>
> ---
> ## [P4] Highlighting technical novelty
> Thank you for your insightful comment. Our work introduces the first automated algorithm for designing hybrid digital twins, with the following key innovations:
> * **Automated design:** We present an evolutionary algorithm that optimizes both model specification and parameterization based on an initial expert prompt. This contrasts with existing hybrid models where experts specify closed-form equations and neural network designs, with only parameters being optimized.
> * **Hybrid model discovery:** Our approach leverages LLMs' capabilities in symbolic manipulation, contextual understanding, and learning to explore a vast combinatorial space of hybrid model specifications. This exploration was previously infeasible with standard techniques, as existing discovery approaches are limited to closed-form equations or neural architecture searches within predefined spaces.
>
> **Actions taken:** Added this to the RWs section to better position our work and showcase its novelty.
>
> ---
> ## Limitations
> We acknowledge the importance of these discussions. **Actions taken:** We have integrated an extended discussion in App A and a summarized version in S8:
> * **Verification of HDTwin models:** First, HDTwins are composed of human-interpretable components, enabling higher degrees of expert verification. Second, evolved models should undergo rigorous functional testing (e.g., held-out datasets, robustness/fairness metrics) pre-deployment. Future works could incorporate hallucination mitigation strategies (e.g., RAG, constitutional AI, multi-model consensus) to further improve reliability.
> * **Variability in evolutionary search:** One strategy lies in quantifying variability through multiple independent runs and confidence intervals. Another lies in controlling variability by providing tighter requirements/constraints in the initial prompt or iterative expert model steering to narrow the hypothesis space.
> * **Ethical implications:** We recognize the potential for bias propagation from black-box LLMs. To address this, we recommend rigorous review verification for fairness, bias, privacy, and other ethical concerns.
>
> ---
> *We hope that most of the reviewer’s concerns have been addressed and, if so, they would consider updating their score. We’d be happy to engage in further discussions.*

---

> > ### Comment · Reviewer_YMit · 2024-08-08
> >
> > The authors have carefully considered my comments and improved the paper accordingly. As a consequence, I'm happy to increase my evaluation.

---

> > > ### Author Response · Authors · 2024-08-09
> > > **Thank you**
> > >
> > > We're happy to have resolved your concerns and are thankful for your input, which has played a key role in refining the quality of our work.

---

### Official Review · Reviewer_UNnB · 2024-07-13

**Soundness:** 2
**Presentation:** 3
**Contribution:** 3
**Rating:** 6
**Confidence:** 3

**Summary:**

An evolutionary search framework for building hybrid dynamics models, especially for so-called digital twins, is proposed. Its notable feature is the use of LLMs for the model proposal and model evaluation in the search. The effectiveness of the method is validated with multiple datasets.

**Strengths:**

- Learning hybrid digital twins is indeed an important research topic.

- The idea of using LLM for the model proposal and evaluation in architecture search is interesting.

- The experimental analysis is done carefully.

**Weaknesses:**

Only a major concern is that in the experiment, methods with (evolutionary or whatever) search but without LLMs are not used. As the use of LLMs in the search is the most notable feature of the method, the experiments should analyze such an aspect particularly. The current baseline methods are okay but do not fit this purpose because they are not based on any search algorithms.

A minor thing. Although the paper nicely overviews some of the relevant studies, it seems to lack a series of studies on hybrid modeling in a part of the ML community, for example:

- Yuan Yin, Vincent Le Guen, Jérémie Dona, Emmanuel de Bézenac, Ibrahim Ayed, Nicolas Thome, and Patrick Gallinari. Augmenting physical models with deep networks for complex dynamics forecasting. Journal of Statistical Mechanics: Theory and Experiment, 2021(12):124012, 2021.
- Naoya Takeishi and Alexandros Kalousis. Physics-integrated variational autoencoders for robust and interpretable generative modeling. Advances in Neural Information Processing Systems 34, pp.14809–14821, 2021.
- Zhaozhi Qian, William. R. Zame, Lucas. M. Fleuren, Paul Elbers, and Mihaela van der Schaar. Integrating expert ODEs into neural ODEs: Pharmacology and disease progression. Advances in Neural Information Processing Systems 34, pp. 11364–11383, 2021.
- Antoine Wehenkel, Jens Behrmann, Hsiang Hsu, Guillermo Sapiro, Gilles Louppe, and Jörn-Henrik Jacobsen. Robust hybrid learning with expert augmentation. Transactions on Machine Learning Research, 2023.

**Questions:**

As noted in the weaknesses section, the lack of baselines using non-LLM search (e.g., a mere evolutionary search) limits the significance of the experimental results. Could you please elaborate on why the authors did not include such baselines? Or, if the current results already imply something in this direction, it would be helpful if the authors could clarify.

**Limitations:**

Limitations clearly discussed.

---

> ### Author Rebuttal · Authors · 2024-08-07
>
> *We thank the reviewer for their thorough review. We are pleased that the reviewer finds our approach interesting, addressing an important research topic, with careful experimental analysis.*
>
> ---
> ## [P1] Incorporating additional baselines
>
> Thank you for this suggestion. While we included SINDy for discovering closed-form governing equations, it doesn't represent an evolutionary search method. To address this, we've added genetic programming (GP) for symbolic regression **[R5]** as a baseline to discover symbolic equations on $\mathcal{D}\_{train}$, using implementation and tuned hyperparameters from **[R6]**.
>
> We have included the additional results in global response **[A1]**, observing that our method discovered superior models. This can be attributed to two key factors:
> 1. **Hybrid model design:** HDTwinGen simultaneously optimizes both the specification (functional form) and parameters of mechanistic and neural components. This hybrid approach enables more flexible and powerful modeling compared to symbolic discovery methods, which are limited to purely symbolic equations.
> 2. **Efficient search:** By leveraging LLMs, HDTwinGen utilizes domain knowledge, contextual understanding, and learning to enhance search efficiency, leading to more accurate solutions.
>
> **Actions taken:** We have incorporated GP-based SR as an additional baseline in our empirical analysis.
>
> ---
> ## [P2] Hyperparameter optimization search
>
> To broaden our comparison with non-LLM methods, we conducted an additional experiment using Bayesian hyperparameter tuning (HPT) search for SINDy and DyNode baselines. For a fair comparison, we matched the number of HPT searches to HDTwinGen's evolutionary search steps (15 precisely). Results presented in **[A3]** demonstrate HDTwinGen's superior performance against these HPT-optimized baselines, highlighting the performance benefits of evolving model specification beyond HPT.
>
> ---
>
> ## [P3] Extending the literature review
>
> We appreciate the references to related works, which consider hybrid models combining mechanistic equations with neural components. **[R1]** integrates prior ODE/PDE knowledge into a hybrid model, using specialized regularization to penalize the neural component's information content. An alternative approach investigates performing hybridization in the latent space. **[R2, R3]** consider settings where an expert equation is known, but equation variables are unobservable. Correspondingly, they employ two sets of latent variables: one governed by expert equations and another linked to neural components. **[R2]** introduced specialized regularization to reduce the divergence of the overall model from the mechanistic component, and semantically grounded expert latent variables. **[R4]** leverages the assumptions that expert models remain valid OOD to sample augmented training distributions from expert latent variables.
>
> Our work differs significantly in both motivation and methodology:
> * **Motivational difference:** Our method introduces an automated approach to jointly optimize hybrid model specification (evolving both mechanistic component and neural architecture) *and* its parameters. This contrasts with related works, where experts specify the hybrid model design and optimization is limited to only its parameters. The benefits of our automated approach are substantial: it enables model development at scale, significantly reducing the time and cost associated with human-driven development. Our method involves humans only at key moments (e.g. providing initial context or iterative feedback, if desired), minimizing efforts required from experts. Importantly, our approach has the potential to uncover novel and effective hybrid model designs that might elude human designers.
> * **Methodological novelty:** Our approach uniquely integrates LLMs within an evolutionary framework to discover optimal models, guided by human-provided task context and data-driven feedback. By leveraging LLMs' advanced capabilities in symbolic manipulation, contextual understanding, and learning, we enable the exploration of a vast combinatorial space of hybrid models. This exploration was previously infeasible with standard evolutionary techniques, marking a significant advancement in automated model discovery and optimization.
>
> **Actions taken:** In response to your suggestion, we have *(1)* extended discussions of related works **[R1-R4]** in L269-280. *(2)* We introduced APHYNITY as an additional baseline in global response **[A1]**, where we observed HDTwinGen outperforming APHYNITY. *(3)* We additionally demonstrated HDTwinGen's ability to further evolve a human-specified APHYNITY model in **[A2]**, highlighting the flexibility of our framework in accommodating various degrees of expert involvement through automated optimization.
>
> ---
> *We hope that most of the reviewer’s concerns have been addressed and, if so, they would consider updating their score. We’d be happy to engage in further discussions.*

---

> > ### Comment · Reviewer_UNnB · 2024-08-13
> >
> > Thank you for the response. The new experiments seem to be informative. I would keep my originally positive score.

---

> > > ### Author Response · Authors · 2024-08-13
> > > **Thank you**
> > >
> > > Thank you for your valuable feedback. We are glad to have addressed your concerns and appreciate your insights, which have significantly enhanced the quality of our work.

---

### Author Rebuttal · Authors · 2024-08-07

*We are grateful to the reviewers for their insightful feedback and constructive comments that have improved the paper.*

We are encouraged by the reviewers' recognition of our work's novelty and potential impact. Reviewers highlighted our approach as "highly novel and of high potential impact" (**8ECc**) with an approach that "integrate domain knowledge by using LLM" (**jD19**) and achieves "significant autonomy in the optimization by means of an evolutionary process" (**YMit**). They noted HDTwinGen as an "advanced algorithm with high potential" that is "general and could be applied to a variety of domains" (**YMit**).

We are pleased that the reviewers agreed that our work addresses "an important research topic" (**UNnb**, **YMit**) and "LLM-driven systems for automating hybrid model design, composition, and optimization are critical" (**8ECc**). Regarding our empirical analysis, reviewers commented, "the analysis is done carefully" (**UNnb**), "outperforming reported baselines" (**jD19**), and "targeted several important desiderata of hybrid models" (**8ECc**).

We have also taken the reviewers’ feedback into account and made the following key changes to improve the paper:

* **[A1] Additional baselines:** We have added comparisons against *(1)* APHYNITY and *(2)* Genetic Programming for symbolic regression.
* **[A2] HDTwinGen optimization of an existing model:** We have provided an additional experiment demonstrating HDTwinGen's ability to further optimize human-specified APHYNITY models used to seed the evolutionary optimization. This highlights the flexibility of our method in accommodating various degrees of expert involvement and its ability to automate the optimization of an existing hybrid modeling technique.
* **[A3] Empirical results with comparable computational budget:** We further compared HDTwinGen against *(1)* DYNODE and *(2)* SINDy with comparable computational resources, allowing 15 iterations of hyperparameter optimization for baselines and 15 iterations of evolutionary model optimization for HDTwinGen.
* **[A4] Extended related works:** Following concrete recommendations, we have expanded the related works section to discuss and compare against existing hybrid models, including Physics-VAE, APHYNITY, LHM, and AHMs.

These revisions have been reflected in the updated manuscript, with additional empirical results provided in the attached PDF.

We believe these updates and our individual responses address the reviewers' concerns and strengthen our paper. We remain open to further feedback.

With thanks,

The Authors of #17161

---

### Additional References

**[R1]** Yin, Y., et al. Augmenting physical models with deep networks for complex dynamics forecasting (2021)

**[R2]** Takeishi, N. and Kalousis, A. Physics-integrated variational autoencoders for robust and interpretable generative modeling, (2021)

**[R3]** Qian, Z., et al. Integrating expert ODEs into neural ODEs: pharmacology and disease progression (2021)

**[R4]** Wehenkel, A., et al. Robust hybrid learning with expert augmentation (2022)

**[R5]** Koza, J.R., Genetic programming as a means for programming computers by natural selection (1994)

**[R6]** Petersen, Brenden K., et al. Deep symbolic regression: Recovering mathematical expressions from data via risk-seeking policy gradients (2020)

**[R7]** Vaghi, C., et al. Population modeling of tumor growth curves and the reduced Gompertz model improve prediction of the age of experimental tumors (2020)

**[R8]** Hong, S., et al. MetaGPT: Meta Programming for A Multi-Agent Collaborative Framework (2024)

**[R9]** Significant-Gravitas. (2023). AutoGPT. GitHub repository. https://github.com/Significant-Gravitas/AutoGPT

---

### Decision · Program_Chairs · 2024-09-25

**Decision:**

Accept (spotlight)

**Comment:**

The paper proposes a neurosymbolic approach to modelling complex dynamical systems, also using domain knowledge and LLMs.

All reviewers recommended acceptance. Reviewers appreciated the novelty of the approach. The authors added baselines in response to reviewers' concerns.


Therefore, I recommend acceptance.